# GO BEYOND YOUR MEANS: UNLEARNING WITH PER-SAMPLE GRADIENT ORTHOGONALIZATION

## ABSTRACT

Machine unlearning aims to remove the influence of problematic training data after a model has been trained. The primary challenge in machine unlearning is ensuring that the process effectively removes specified data without compromising the model's overall performance on the remaining dataset. Many existing machine unlearning methods address this challenge by carefully balancing gradient ascent on the 'unlearn' data with the gradient descent on a 'retain' set that represents the training data. However, in many cases the training dataset is not fully available when we wish to unlearn some concepts, because models are released without their training datasets, and one may only have access to a *small part of a training set*. Here, we propose OrthoGrad, a novel approach that mitigates interference between the unlearn set and a small retain set rather than competing ascent and descent processes. Our method projects the gradient of the unlearn set onto the subspace orthogonal to all gradients in the retain batch, effectively avoiding any gradient interference. We demonstrate the effectiveness of OrthoGrad on multiple machine unlearning benchmarks, including automatic speech recognition, outperforming competing methods.

## 1 INTRODUCTION

Foundation models are trained on web-scale datasets, which may contain undesirable data: illegal, proprietary, or privacy-infringing. For example, Github Copilot (Dakhel et al., 2023; Siroš et al., 2024) faced criticism for generating code snippets directly from open-source repositories without attribution, and the LAION-5B dataset (Schuhmann et al., 2022) had to be temporarily removed when it was discovered it contained CSAM images (Thiel, 2023). Another type of undesirable data is the case were users may ask to 'opt out' and to not be recognized by the system. For example, a user might want a speech recognition system to not transcribe his audio recordings. In all these cases, one is interested to "remove" or "forget" information from a pre-trained model, either general knowledge or specific information.

These challenges led to a growing recent interest in *machine unlearning* (Liu et al., 2024; Nguyen et al., 2022). In this setup, we wish to remove the effects of a given part of the training data on a pretrained model while preserving its generalization performance. In practice, we are given an *unlearn set* that we wish to forget and a *retain set* that represents the training data. Many existing methods (Kurmanji et al., 2024; Lin et al., 2024) combine gradient ascent on an **unlearn set** – for degrading performance on selected data, with gradient descent on a **retain set** – for preserving accuracy elsewhere.

Very often however, models are released without their full training dataset, and one may only have access a small fraction of the training data to serve as a retain set. For instance, Whisper large-V3 (Radford et al., 2023), an ASR foundation model, was trained on a proprietary dataset comprising over 5 million hours of labeled audio recordings. Although this private dataset cannot serve as a retain set, small-scale publicly available ASR datasets such as LibriSpeech can be used as substitutes. The key observation of this paper is that leading unlearning methods average over the retain set. However, when the retain set is small, one aims to go beyond averages and extract as much information as possible from the retain set.

In this work, we tackle the challenge of machine unlearning with a limited retain set. We propose a novel algorithm named *OrthoGrad*, which enables effective unlearning while minimizing the impact

on the model's generalization performance. The key idea is to use the gradients over the retain set to estimate a subspace of gradients that should be maintained. This way, rather than relying heavily on the retain set to offset the negative effects of the unlearning process, our method directly mitigate interference by taking update steps that are orthogonal to the retain subspace.

To motivate our approach, we begin with a theoretical analysis under simplifying assumptions. The ideal objective of unlearning is to modify performance on the unlearn set while preserving performance on the retain set. This can be framed as an optimization problem constrained to the manifold of parameters that leave all retain-set points unaffected. We show that the gradient restricted to this manifold is equivalent to projecting the unlearning gradient onto the subspace orthogonal to the per-sample gradients of the retain batch. Building on this insight, we develop an algorithm that efficiently approximates the corresponding optimization trajectory. Unlike prior methods that rely on the average retain-set gradient, our approach adopts a *per-sample gradient* perspective, yielding a more robust solution to unlearning (Figure 1).

Our experiments focus on the challenging regime of small retain sets. These settings highlight the practical constraints often encountered in real-world applications of machine unlearning. We thoroughly evaluate the effectiveness of our approach, OrthoGrad, across several challenging tasks, including image classification and automatic speech recognition. Additionally, we evaluate our approach across diverse unlearning regimes, including random data removal, class-specific forgetting, and a proxy-retain setting where the retain set is drawn from a related but distinct distribution, demonstrating versatility when the original training data are unavailable. Our results consistently show that OrthoGrad achieves reliable unlearning while maintaining the overall model performance better than other leading unlearning methods.

This paper makes the following contributions: (i) We propose *OrthoGrad* – a new machine unlearning method, tailored for a limited amount of retain data. (ii) From a geometric perspective, we provide a theoretical motivation for our approach. (iii) We demonstrate the effectiveness of OrthoGrad through extensive experiments spanning multiple datasets, modalities, and unlearning setups.

## 2 RELATED WORK

The development of efficient machine unlearning methods (Cao & Yang, 2015; Fan et al., 2025; Ginart et al., 2019; Goel et al., 2022; Zhang et al., 2024; Romero et al., 2007; Mehta et al., 2022; Huang et al., 2025) has gained significant attention, addressing a range of applications across domains such as regression tasks (Thudi et al., 2022), federated learning (Liu et al., 2021b; 2022; Wang et al., 2022), graph neural network (Chen et al., 2022; Cheng et al., 2023). Retraining the model from scratch, widely regarded as the gold standard for unlearning Fan et al. (2023), guarantees the complete removal of data influence. However, this approach is often impractical in production environments due to the extensive computational resources, especially for large-scale datasets. Alterna-

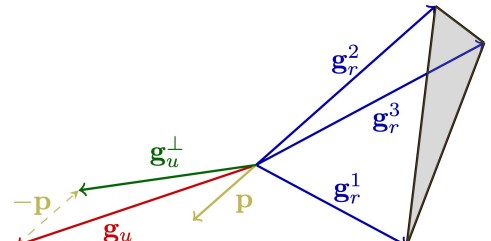

Figure 1: Illustration of the gradient orthogonalization process. The retain gradients $\mathbf{g}_r^1$, $\mathbf{g}_r^2$, and $\mathbf{g}_r^3$ span a subspace (gray triangle). The projection vector $\mathbf{p}$ is obtained by applying QR decomposition on the retain gradients. The unlearn gradient $\mathbf{g}_u$ is projected using $\mathbf{p}$ to form unlearning gradient which is orthogonal to the retain subspace, $\mathbf{g}_u^{\perp}$.

tively, fine-tuning a model for a new task may induce catastrophic forgetting (Lopez-Paz & Ranzato, 2017), but this mechanism fails to ensure the precise removal of specific data influences.

Most machine unlearning methods leverage techniques like influence functions (Guo et al., 2019; Neel et al., 2021; Wu et al., 2022; Wu & Harandi, 2025; Sekhari et al., 2021), probabilistic approaches (Golatkar et al., 2020b; 2021). However, these methods often face inherent limitations that reduce their practical effectiveness, particularly in defending against membership inference attacks (Dwork et al., 2006; Graves et al., 2021). As a result, the focus has shifted toward developing more effective and efficient unlearning strategies (Golatkar et al., 2020a; Becker & Liebig, 2022; Jia et al., 2023; Chen et al., 2023). While these approaches represent significant advancements in machine unlearning, many rely on assumptions or techniques that limit their practicality in real-world scenarios.

Cluster-based unlearning (DUCK (Cotogni et al., 2023), SCAR (Bonato et al., 2024)), differing mainly in clustering metrics, lack adaptation to auto-regressive models, limiting applicability to sequential tasks. SCRUB (Kurmanji et al., 2024), a teacher–student framework, removes specific influences but struggles to generalize (e.g., forgetting random samples). GDR-GMA (Lin et al., 2024) relies on orthogonal projections of averaged gradients, ignoring per-sample variability and leaving residual influence. Most methods target classification; (Fan et al., 2023) highlights limitations for image generation, crucial for copyright and safety.

**Conflicting Gradients in Multi-Task Learning:** Multi-task learning (MTL) aims to improve model generalization by optimizing multiple related tasks (Crawshaw, 2020; Zhang & Yang, 2021). However, different tasks often compete for model capacity and produce gradients pointing in opposite directions during training (Yu et al., 2020). This phenomenon, known as gradient interference or conflict, occurs when a gradient that benefits one task degrades the performance of others. Addressing this issue by mitigating these conflicts has become crucial for training MTL systems, with early works focusing on analyzing conflict patterns and their relationships (Sener & Koltun, 2018; Chen et al., 2018). Various optimization-based approaches have been proposed, including gradient projection and dropping to reduce task interference (Chen et al., 2020; Wang et al., 2021; Liu et al., 2021a; Achituve et al., 2024). Recently, geometric and game-theory perspectives have led to methods seeking optimal Pareto solutions in the MTL optimization landscape (Navon et al., 2022; Javaloy & Valera, 2021). Other studies proposed architectural solutions, including progressive networks (Rusu et al., 2016), attention-based routing (Ma et al., 2019), and dynamic architecture adaptation (Sun et al., 2020). Another line of works focuses on dynamic loss weighting, with methods like uncertainty weighting (Kendall et al., 2018) and DWA (Liu et al., 2019) automatically balancing task losses based on pre-defined criteria. In the related field of continual learning, Zeng et al. (2019) proposed enforcing orthogonality between gradient vectors to mitigate catastrophic forgetting. These advances in handling conflicting gradients provide valuable insights for machine unlearning, where the goal is to satisfy both the unlearning objective and maintain performance on retained data.

In this work, we focus on evaluating machine unlearning across various setups in the contexts of image classification and Automatic Speech Recognition (ASR). By exploring different unlearning scenarios, we aim to test the generalization capabilities of all methods, particularly in handling large-scale datasets and scenarios with restricted access to the original training data.

## 3 BACKGROUND

We consider a training dataset $\mathcal{D} = \{(x_i, y_i)\}_{i=1}^{N}$ with each data point representing a pair of input vector $x_i$ and its corresponding label $y_i$. A machine learning model $f(\cdot; \theta)$, parameterized by parameters $\theta$, is optimized to minimize a loss function $\mathcal{L}(\theta)$. Formally, we define the loss function as $\mathcal{L}(\theta) = \frac{1}{N} \sum_{i=1}^{N} \ell(f(x_i; \theta), y_i)$, where $\ell$ is the cross-entropy loss. The model parameters trained on $\mathcal{D}$ are denoted as $\theta_p$ representing the *pretrained model*. In machine unlearning, we are given two datasets: (i) Unlearn set $\mathcal{D}_u = \{(x_i, y_i)\}_{i=1}^{N_u}$, containing the $N_u$ data points to be unlearned. (ii) Retain set $\mathcal{D}_r = \{(x_i, y_i)\}_{i=1}^{N_r}$, with $N_r$ samples representing the training data to aid retain the model's performance. We assume these two datasets are disjoint, i.e., $\mathcal{D}_u \cap \mathcal{D}_r = \emptyset$. The primary goal of machine unlearning is to modify the model's weights to obtain $\theta_u$ resulting in an *unlearned model* $f(\cdot; \theta_u)$. This modification process aims to remove the knowledge of the original model of $\mathcal{D}_u$. At the same time, the model must maintain its predictive performance on unseen data.

One major challenge in machine unlearning is how to define if the unlearn set was successfully unlearned. While theoretically, we want our model to be indistinguishable from a model trained from scratch without the unlearn set, this is hard to verify without actually retraining from scratch. In this work, similar to Cotogni et al. (2023), we aim for the performance on the unlearn set to match the original models' performance on the test set as a proxy. This also has the added benefit of making the comparison between different unlearning methods straightforward. As we care about both unlearn set performance and test set performance, by normalizing the unlearn set performance of all models (up to some tolerance), we can directly compare using a single metric, i.e., the test set performance.

# 4 METHOD

We introduce OrthoGrad, a novel machine unlearning approach designed to address unlearning with *limited retain data*. We observe that current unlearning methods perform gradient ascent for unlearning and gradient descent for retention. Such approaches excessively depend on the retain set, because in a sense, we are simultaneously forgetting and retraining during the unlearning phase. Such mixed objectives are known to be harder to stabilize and optimize. Considering this, our approach aims to mitigate the negative effect of the unlearning step instead of fixing it using the retain set. We propose to use a small retain set in a more efficient way, by computing a subspace of gradients that should not interfere with the retained data.

## 4.1 GEOMETRIC MOTIVATION

To motivate our method, we start by analyzing theoretically how we would perform ideal unlearning under strong simplifying assumptions. This will then guide the design of our practical algorithm. Intuitively, we are interested in the set of parameter vectors $\theta \in \mathbb{R}^d$ that maintains a constant loss over the retain set $\mathcal{D}_r$. Formally, let $\bar{\mathcal{L}}_r(\theta) = (\ell_1(\theta), ..., \ell_{N_r}(\theta))$ define the vector of losses over the retain set, we are interested in performing unlearning in the level set $\bar{\Theta} := \{\theta \in \mathbb{R}^d \mid \bar{\mathcal{L}}_r(\theta) = \bar{\mathcal{L}}_r(\theta_p)\}$, i.e., unlearning without changing the loss on elements of the retain set.

**Claim 1.** *Assuming that (i) The loss $\ell$ is continuously differentiable, and (ii) The Jacobian of the retain loss $\nabla \bar{\mathcal{L}}_r \in \mathbb{R}^{N_r \times d}$ is of full rank for all $\theta \in \bar{\Theta}$ then $\bar{\Theta}$ is a smooth manifold of dimension $d - N_r$*

**Proof Sketch.** To see that, we define for $\theta' \in \bar{\Theta}$ the function $f : \mathbb{R}^d \to \mathbb{R}^{N_r}$ by $f(\theta) = \bar{\mathcal{L}}_r(\theta) - \bar{\mathcal{L}}_r(\theta_p)$. From our assumptions, $f$ is continuously differentiable, and the Jacobian of $f$, $\nabla f = \nabla \bar{\mathcal{L}}_r$ has full rank at $\theta'$. Thus, as a direct result of the implicit function theorem, the set $\bar{\Theta}$ is locally diffeomorphic to an open ball in $\mathbb{R}^{d-N_r}$. $\square$

Considering continuous parameter updates, to minimize the unlearn loss while remaining in $\bar{\Theta}$, we follow the gradient flow restricted to the manifold. This requires projecting the Euclidean gradient of our objective onto the tangent space $T_\theta \bar{\Theta}$, ensuring the flow stays on the manifold.

**Claim 2.** *The tangent space $T_{\theta'} \bar{\Theta}$ to $\bar{\Theta}$ at $\theta'$ is given by the null space of the Jacobian $\nabla_{\theta'} \bar{\mathcal{L}}_r$, that is, the set of directions in parameter space that are orthogonal to the subspace spanned by the retain gradients, $T_{\theta'} \bar{\Theta} = \{v \in \mathbb{R}^d \mid \nabla \bar{\mathcal{L}}_r v = 0\}$*

**Proof Sketch.** To show that $T_{\theta'} \bar{\Theta} = \text{Ker}(\nabla_{\theta'} \bar{\mathcal{L}}_r)$, we first note that $T_{\theta'} \bar{\Theta} \subset \text{Ker}(\nabla_{\theta'} \bar{\mathcal{L}}_r)$: Let $v \in T_{\theta'} \bar{\Theta}$ and let $\gamma(\cdot)$ be a smooth curve in $\bar{\Theta}$ with $\gamma(0) = \theta'$ and $\dot{\gamma}(0) = v$, we have $0 = \frac{d}{dt} f(\gamma(t)) \mid_{t=0} = \nabla_{\theta'} \bar{\mathcal{L}}_r v$, and so $v \in \text{Ker}(\nabla_{\theta'} \bar{\mathcal{L}}_r)$. Finally, we get the equality $T_{\theta'} \bar{\Theta} = \text{Ker}(\nabla_{\theta'} \bar{\mathcal{L}}_r)$ following a dimension counting argument since $\dim(T_{\theta'} \bar{\Theta}) = \dim(\text{Ker}(\nabla_{\theta'} \bar{\mathcal{L}}_r)) = d - N_r$. $\square$

To algorithmically perform this gradient flow we would need to compute the standard gradient, project it to the space orthogonal to the gradients of the entire retain set, and then update the parameters along the exponential map, or update and then project back to the manifold. This, however, is very demanding computationally, as we need to compute and store the gradients on the entire retain set, as well as compute the exponential map or projection step.

## 4.2 PRACTICAL ALGORITHM

While performing the exact gradient flow on the retain set is too computationally expensive to run in practice, it inspires the design of our simple and practical unlearning algorithm OrthoGrad. At each optimization step, we simply project the unlearn gradient to the space orthogonal to all individual gradients of the retain batch. Specifically, at each step, we sample a batch of examples from the unlearn set $\mathcal{D}_u$ and calculate the mean gradient vector on this batch. We denote this gradient vector as $g_u$. Next, we sample a batch from the retain set $\mathcal{D}_r$. For the retain batch with $k$ samples, we compute the per-sample gradient matrix $G_r = [g_r^1, g_r^2, \ldots, g_r^k]$, where each column $g_r^i$ corresponds to the gradient vector for sample $i$ in the batch. Importantly, we note that this can be achieved efficiently using modern automatic differentiation libraries, such as PyTorch (Paszke et al., 2019), which allow us to obtain per-sample gradients in a single forward-backward pass. To ensure orthogonality between $g_u$ and the column space of $G_r$, we employ QR decomposition (Francis, 1961) on $G_r$. This yields an orthonormal basis $Q = [q_r^1, q_r^2, \ldots, q_r^k]$ that spans this subspace. Once

the retain gradient subspace is defined, we project the unlearn gradient onto this subspace to compute its projection w.r.t each subspace vector. For a single retain gradient $g_r^i$, the projection is calculated as: $g_u^\perp = g_u - \sum_{i=1}^k \langle g_u, q_r^i \rangle q_r^i$. We note that while previous algorithms, for example Lin et al. (2024), do try to mitigate interference between the unlearn and the retain set gradients, they achieve this on the batch-level average gradients and not on the gradients of the individual data points. We found in our experiments that the more strict per-element constraint, instead of working on the mean gradient gives a stronger performance (Section 5.1.1).

We now discuss two modifications of our method that we found to provide large empirical gains. First, instead of changing the entire weight space, we use low-rank adaptation (LoRA) (Hu et al., 2021) to limit further the effect that unlearning has on the overall test performance. We note that in general parameter parameter-efficient fine-tuning (PEFT) is a rapidly evolving field, and how to utilize it for unlearning best have yet to be thoroughly explored. Second, while our method is robust to retain-set size, it may underuse it; linearly combining retain and unlearn gradients improves the performance of the unlearned model compared to solely performing gradient ascent in the direction of the unlearn gradient. Therefore, we define the update gradient as: $g = \alpha \bar{g}_r - (1 - \alpha) g_u^\perp$ where $\bar{g}_r = \frac{1}{k} \sum_{i=1}^k g_r^i$ is the retain gradient averaged over the batch, and $\alpha \in [0, 1]$ is a hyperparameter that controls the trade-off between forgetting and retaining. Finally, we update the model parameters $\theta$ using the update rule: $\theta_l \leftarrow \theta_l - \eta g$. The step-by-step procedure is presented in Algorithm 1.

In summary, OrthoGrad enforces orthogonality between the unlearn and retain gradients, minimizing the interference between the updates of the unlearn set and retain set. OrthoGrad is designed for low-data regimes (small retain sets), because unlike previous methods, it takes into account the subspace of gradients defined by the retain set, rather than average aggregates only. We demonstrate OrthoGrad effectiveness on various datasets and model architectures in the next section.

## 5 EXPERIMENTS

We evaluate OrthoGrad and compare it with recent machine unlearning approaches. We use several datasets, model architectures, and unlearning setups to demonstrate the effectiveness and versatility of OrthoGrad in the regime of a limited number of retain data points. To encourage future research and reproducibility, we will make our code publicly available. Additional experimental results are presented in Appendix A, including insightful analyses, ablation studies on key hyperparameters, and a detailed discussion of evaluation metrics.

**Baselines.** We compare OrthoGrad with recent machine unlearning baselines. (1) Retrain - retraining from scratch without the unlearn set. *We note that this baseline is inappropriate in the low data regime since it overfits the retain data, but we include it for completeness.* (2) Finetune - finetune the pretrained model solely with the retain set. (3) NegGrad (Graves et al., 2021; Thudi et al., 2022) - a naive approach that performs gradient ascent steps on the unlearn set. (4) NegGrad+ (Kurmanji et al., 2024) - NegGrad with the additional goal of minimizing retain loss and preserving the model's knowledge on the retain dataset. (5) FISHER (Golatkar et al., 2020a) - adds additive noise to the pretrained weights with a constraint on the fisher information matrix. (6) Influence (Koh & Liang, 2017; Izzo et al., 2021) - utilizes influence functions to identify the parameters most critical to the data being unlearned and perturb them by adding additive noise. (7) SCRUB (Kurmanji et al., 2024) - A knowledge distillation approach that incorporates a regularization term into the unlearning objective. (8) DUCK (Cotogni et al., 2023) - uses metric learning to minimize the distance between feature vectors of the data to be forgotten and the nearest centroid of a different class. (9) SCAR (Bonato et al., 2024) - similar to DUCK, it uses Mahalanobis distance as the objective to minimize. (10) SSD (Foster et al., 2024) - uses Fisher information to identify parameters tied to the forget set and selectively dampens them. We note that SCAR and SSD rely less on the retain set. (11) GDR-GMA (Lin et al., 2024) - projecting conflicting gradients onto an orthonormal plane and dynamically adjusting the magnitude of update gradients.

**Evaluation.** We report the two common evaluation metrics in the field: (1) unlearning accuracy ($\mathcal{A}_u$) on the data to be forgotten, and (2) test accuracy ($\mathcal{A}_{test}$) on the held-out test set. For completeness, we also report retain accuracy ($\mathcal{A}_r$) on the retain data. This is comparable to train accuracy in standard learning and should not be used for comparison. In all experiments, we perform early stopping based

on $\mathcal{A}_u$ reaching a specific target (normally the original test accuracy). This is easier to compare because the main difference is in $\mathcal{A}_{test}$. Stopping criteria are crucial in machine unlearning to ensure the process reaches a proper balance between effective forgetting with retained functionality.

As machine unlearning involves multiple objectives, we propose the following *Unlearning Impact Score* (UIS) for easier comparison. Our metric is defined as:

$$ UIS = \left( \frac{|\mathcal{A}_{test}^p - \mathcal{A}_{test}^u|}{\mathcal{A}_{test}^p} + \frac{|\mathcal{A}_{test}^p - \mathcal{A}_u^u|}{\mathcal{A}_{test}^p} \right) / 2 \quad , $$

where the up scripts $p$ and $u$ denote pretrained and unlearned models respectively. In UIS we average two components: the relative change in test accuracy, and how close the performance on the unlearning set is to its target, $\mathcal{A}_{test}^p$. A lower UIS score indicates better unlearning, as it suggests the model has successfully forgotten the unlearn data while maintaining its performance on held-out data. Additional results with the MIA metric are in the appendix.

## 5.1 AUTOMATIC SPEECH RECOGNITION

Automatic Speech Recognition (ASR) is the process of converting spoken language into written text, a fundamental component in many real-world applications (Malik et al., 2021; Alharbi et al., 2021). ASR foundation models like Whisper (Radford et al., 2023) are trained on extensive datasets of transcribed web audio containing many hours of speech recordings. These models may inadvertently retain sensitive or proprietary information. Furthermore, individuals may request that an ASR system cannot accurately transcribe their voice as a way to preserve their privacy and identity.

**Speaker unlearning.** We focus on the task of forgetting audio data associated with a particular speaker, using Whisper-Tiny (Radford et al., 2023) architecture and LibriSpeech (Panayotov et al., 2015) dataset, containing 1K hours of English speech recordings. We establish the unlearning setup by selecting a single speaker from the training set to serve as the unlearn set. We randomly allocate 10% from the unlearn set to evaluate our model on the unlearned speaker. Additionally, we randomly sample 10% of the remaining training set to form the retain set. The test set is taken directly from the original LibriSpeech dataset.

**Eval metrics.** We evaluate performance using word error rate (WER), a standard metric that measures the percentage of words incorrectly transcribed by the model. We report WER for 4 sets: unlearn ($\mathcal{W}_{unlearn}$), retain ($\mathcal{W}_{retain}$), test ($\mathcal{W}_{test}$), and speaker held out ($\mathcal{W}_{speaker}$). The speaker held-out dataset comprises of unseen audio recordings of the unlearned speaker. Since Whisper tends to hallucinate (Koenecke et al., 2024) by predicting unwanted words, we clip the WER at a maximum of 100%.

### 5.1.1 ABLATION STUDY

We begin with an ablation study to evaluate the relative contribution of each component in our approach. We run the unlearning process for 30 epochs with an early stopping when $\mathcal{W}_{unlearn}$ reaches 75%. WER tends to jump significantly during the last epochs, which can lead to a final WER that is much higher than our stopping criteria. Although the exact threshold is somewhat arbitrary, we observed a rapid increase in WER beyond a certain point. This suggests that the results should remain robust regardless of the specific threshold chosen.

In this experiment, we compare 5 variants. (i) *OrthoGrad Mean*; Projecting the unlearn gradient to be orthogonal to the average retain gradient, (ii) *OrthoGrad Per-sample*; Projecting the unlearn gradient

Table 1: *Ablation Study.* Evaluation of OrthoGrad variants on ASR unlearning. Values are word-error-rates averaged over 5 different speakers.

|  | $\mathcal{W}_{retain}$ | $\mathcal{W}_{unlearn}$ | $\mathcal{W}_{speaker}$ | $\mathcal{W}_{test}$ |
|---|---|---|---|---|
| OrthoGrad Mean | $27.23 \pm 11.36$ | $96.67 \pm 6.02$ | $64.25 \pm 35.48$ | $29.42 \pm 14.07$ |
| OrthoGrad Per-sample | $18.71 \pm 4.04$ | $100.00 \pm 0.00$ | $96.40 \pm 7.04$ | $26.87 \pm 0.60$ |
| OrthoGrad Mean + Lora | $23.77 \pm 9.62$ | $92.12 \pm 7.34$ | $63.27 \pm 35.43$ | $41.21 \pm 25.67$ |
| OrthoGrad Per-sample + Lora | $12.73 \pm 1.43$ | $98.30 \pm 2.50$ | $81.16 \pm 23.97$ | $16.36 \pm 0.32$ |
| OrthoGrad | $12.11 \pm 0.65$ | $96.24 \pm 8.06$ | $98.53 \pm 3.28$ | $\mathbf{13.98 \pm 0.58}$ |

to the space orthogonal to all individual sample gradients, (iii) *OrthoGrad Mean/Per-sample+LoRA*; The latter methods when the update is restricted to low-rank adapters. (iv) *OrthoGrad*; Our full method that combines gradient descent on the retain set.

Table 1 shows the results. Per-sample orthogonalization has two benefits. It reduces the mean Word-Error-Rate $\mathcal{W}_{test}$ and also reduces its variance by an order of magnitude. We observed that OrthoGrad Mean is very unstable: it may work well with some speakers but performs poorly on others. As seen in Table 1, restricting per-sample unlearning of OrthoGrad to LoRA adapters improves $\mathcal{W}_{test}$ significantly. However, this is not the case for OrthoGrad Mean due to instability. We note that all methods passed the $75\%$ WER threshold on the unlearn set with a large margin, but the OrthoGrad Mean performance on unseen audio from the speaker, $\mathcal{W}_{speaker}$, was below the target threshold. This means the unlearning did not generalize well to new recordings of the unlearned speaker. Finally, we see that adding the retain gradient can offer an additional improvement, but this improvement is somewhat limited.

### 5.1.2 ASR Speaker Unlearning Results

For speaker unlearning, we compare OrthoGrad to SCRUB Kurmanji et al. (2024), GDR-GMA Lin et al. (2024), and NegGrad+ Kurmanji et al. (2024). We exclude metric learning methods (DUCK and SCAR) as they are designed for classification and are unsuitable for ASR. Also, SSD relies on trained parameters, making it unsuitable for optimizing LoRA in this unlearning setup. See technical details and hyperparameter selection in Appendix B.2.

The results are shown in Table 2. All methods, except for the finetune baseline, successfully unlearned the target speaker. We hypothesize that the high $\mathcal{W}_{test}$ values for both NegGrad+ and SCRUB arise from the fact that they do not take into account the conflict between the unlearn and retain gradients. In contrast, OrthoGrad and GDR-GMA, which consider this conflict, perform well on this benchmark. OrthoGrad significantly outperforms GDR-GMA, on test WER.

### 5.2 Unlearning with Proxy Data

In practice, the original training data are usually unavailable, especially for foundation models trained on copyrighted or proprietary data. As a result, practitioners who wish to perform unlearning must curate a small proxy retain set that approximates the original data distribution. To simulate this scenario, we evaluate OrthoGrad in a proxy-retain setting using CINIC-10 (Darlow et al., 2018), which merges CIFAR-10 with resized ImageNet images from the same classes. We first train a ResNet-18 on CIFAR-10. During unlearning, we construct the retain set exclusively from the ImageNet-derived portion of CINIC-10, uniformly sampling 10% of this pool. We use CIFAR-10 examples as both the forget set and the test set. This protocol enforces a distribution shift between retain and forget/test while preventing any retain-set leakage from CIFAR-10. Results are reported in Table 3.

OrthoGrad achieves the lowest UIS in both random-sampling and class-forgetting, lowering $A_u$ while keeping $A_{\text{test}}$ near the pretrained model despite the proxy (distribution-shifted) retain set. In contrast, baselines leave residual memorization (high $A_u$) or cause large drops in $A_r$ or $A_{\text{test}}$. Other methods effectively fail to unlearn; for example, SSD did not achieve any unlearning in the random-forgetting

Table 2: *Automatic Speech Recognition.* ASR speaker unlearning results on the LibriSpeech dataset. Values are word-error-rates averaged over 5 different speakers.

| Method | $\mathcal{W}_{retain}$ | $\mathcal{W}_{unlearn}$ | $\mathcal{W}_{speaker}$ | $\mathcal{W}_{test}$ |
|---|---|---|---|---|
| Original | $9.99 \pm 0.15$ | $11.12 \pm 4.91$ | $10.06 \pm 6.39$ | $11.08 \pm 0.00$ |
| Finetune | $0.06 \pm 0.01$ | $13.39 \pm 5.26$ | $12.54 \pm 7.48$ | $13.67 \pm 0.04$ |
| NegGrad+ | $72.87 \pm 19.18$ | $77.08 \pm 35.99$ | $\underline{94.89 \pm 6.78}$ | $85.90 \pm 10.72$ |
| SCRUB | $100.00 \pm 0.00$ | $100.00 \pm 0.00$ | $100.00 \pm 0.00$ | $100.00 \pm 0.00$ |
| GDR-GMA | $17.38 \pm 9.69$ | $93.28 \pm 7.58$ | $94.76 \pm 6.23$ | $\underline{32.52 \pm 5.72}$ |
| OrthoGrad | $12.11 \pm 0.65$ | $96.24 \pm 8.06$ | $\mathbf{98.53 \pm 3.28}$ | $\mathbf{13.98 \pm 0.58}$ |

Table 3: *Proxy-Retain with ResNet18 architecture.* Performance is measured under two unlearning scenarios: random sampling of training data (3-seed average) and class removal (3-class average).

| Method | Random Sampling | | | | Class Forgetting | | | |
|---|---|---|---|---|---|---|---|---|
| | $\mathcal{A}_u$ | $\mathcal{A}_r$ | $\mathcal{A}_{test}$ | UIS ($\downarrow$) | $\mathcal{A}_u$ | $\mathcal{A}_r$ | $\mathcal{A}_{test}$ | UIS ($\downarrow$) |
| Original | $96.10 \pm 0.28$ | $49.45 \pm 1.09$ | $81.97 \pm 0.00$ | – | $97.31 \pm 1.21$ | $48.55 \pm 1.71$ | $81.97 \pm 0.00$ | – |
| Retrain | $29.95 \pm 3.11$ | $99.70 \pm 0.43$ | $30.48 \pm 2.65$ | – | $0.00 \pm 0.00$ | $99.94 \pm 0.09$ | $28.18 \pm 1.94$ | – |
| FT | $78.61 \pm 1.38$ | $72.30 \pm 0.56$ | $67.78 \pm 1.17$ | – | $29.23 \pm 14.15$ | $99.95 \pm 0.05$ | $64.26 \pm 0.45$ | – |
| NegGrad | $43.63 \pm 32.62$ | $24.30 \pm 13.96$ | $37.24 \pm 26.17$ | $0.507 \pm 0.359$ | $0.27 \pm 0.46$ | $20.43 \pm 10.87$ | $29.41 \pm 21.75$ | $0.322 \pm 0.130$ |
| NegGrad+ | $21.43 \pm 2.64$ | $28.36 \pm 1.67$ | $19.89 \pm 1.95$ | $0.748 \pm 0.028$ | $0.76 \pm 0.45$ | $42.07 \pm 4.62$ | $51.15 \pm 13.24$ | $0.193 \pm 0.080$ |
| FISHER | $10.53 \pm 0.60$ | $10.28 \pm 0.34$ | $10.18 \pm 0.30$ | $0.874 \pm 0.005$ | $71.15 \pm 32.00$ | $39.01 \pm 2.03$ | $63.81 \pm 0.49$ | $0.545 \pm 0.196$ |
| Influence | $10.22 \pm 0.55$ | $10.09 \pm 0.10$ | $10.00 \pm 0.00$ | $0.877 \pm 0.003$ | $72.01 \pm 32.84$ | $41.43 \pm 5.55$ | $73.97 \pm 7.63$ | $0.488 \pm 0.154$ |
| SCRUB | $40.21 \pm 6.51$ | $42.56 \pm 5.88$ | $38.63 \pm 4.30$ | $0.519 \pm 0.066$ | $1.20 \pm 1.19$ | $64.81 \pm 1.02$ | $52.36 \pm 2.05$ | $0.188 \pm 0.019$ |
| DUCK | $53.22 \pm 5.82$ | $99.47 \pm 0.18$ | $46.43 \pm 4.01$ | $0.392 \pm 0.060$ | $0.00 \pm 0.00$ | $42.53 \pm 2.76$ | $22.94 \pm 6.53$ | $0.360 \pm 0.040$ |
| GDR-GMA | $79.93 \pm 0.70$ | $93.97 \pm 0.89$ | $67.02 \pm 0.45$ | $0.104 \pm 0.007$ | $0.00 \pm 0.00$ | $51.74 \pm 7.53$ | $56.69 \pm 6.11$ | $0.154 \pm 0.037$ |
| SSD | - | - | - | - | $96.59 \pm 2.14$ | $48.12 \pm 0.95$ | $81.10 \pm 1.57$ | $0.595 \pm 0.005$ |
| SCAR | $78.05 \pm 4.29$ | $42.79 \pm 2.29$ | $68.32 \pm 3.60$ | $0.107 \pm 0.048$ | $0.09 \pm 0.09$ | $48.74 \pm 0.80$ | $64.44 \pm 6.22$ | $0.107 \pm 0.037$ |
| OrthoGrad | $80.99 \pm 1.21$ | $61.69 \pm 0.71$ | $68.41 \pm 0.79$ | $\mathbf{0.089 \pm 0.012}$ | $0.46 \pm 0.42$ | $59.58 \pm 1.36$ | $68.94 \pm 3.81$ | $\mathbf{0.082 \pm 0.021}$ |

setup, even after hyperparameter tuning. These results suggest that orthogonalizing updates to the retain gradient subspace provides better unlearning with scarce proxy data.

## 5.3 IMAGE CLASSIFICATION

Image classification tasks are commonly used benchmarks for evaluating machine unlearning algorithms. These benchmarks have two variations: class-wise forgetting and random data forgetting. Class-wise forgetting focuses on removing the influence of an entire image class, while random data forgetting targets the removal of randomly selected data points from the training set. In the standard experimental setup, the entire training set, except for the unlearn set, is used as the retain set. We, however, are interested in the scenario where we have access to a limited retain set, and therefore subsample a portion of the training set to be our retain set.

Our evaluation is conducted on the ImageNet (Deng et al., 2009) image classification dataset on both random sampling and class forgetting benchmarks. In the random unlearning setting, the unlearn set consists of 5K images sampled uniformly from the training data. In the class unlearning setting, the unlearn set comprises all training images belonging to the unlearn class. In both setups, we draw 10K images for the retain set and evaluate on the original test set. We use ResNet-18 (He et al., 2016) and ViT (Dosovitskiy, 2020) as our base classifiers. The stopping criteria used in the random forgetting experiments follow Cotogni et al. (2023), i.e., we stop when the unlearn accuracy is within a defined threshold (0.5%) or lower than the test accuracy of the pretrained model. For class-wise forgetting, we stopped when the model's accuracy on the unlearned classes dropped below 1%, indicating that the class had been effectively forgotten. In both cases, if the unlearning algorithm does not reach the target within a specific number of epochs, we report the results of the last epoch.

Table 4: *ImageNet using ResNet18.* Results are shown for two scenarios: random sampling (averaged over 3 seeds) and class forgetting (averaged over 3 classes).

| Method | Random Sampling | | | | Class Forgetting | | | |
|---|---|---|---|---|---|---|---|---|
| | $\mathcal{A}_u$ | $\mathcal{A}_r$ | $\mathcal{A}_{test}$ | UIS ($\downarrow$) | $\mathcal{A}_u$ | $\mathcal{A}_r$ | $\mathcal{A}_{test}$ | UIS ($\downarrow$) |
| Original | $79.04 \pm 0.68$ | $79.2 \pm 0.49$ | $69.76 \pm 0.00$ | – | $91.76 \pm 4.79$ | $79.15 \pm 0.00$ | $69.76 \pm 0.00$ | – |
| Retrain | $5.72 \pm 0.16$ | $95.90 \pm 0.32$ | $5.72 \pm 0.35$ | – | $0.00 \pm 0.00$ | $84.60 \pm 1.35$ | $5.34 \pm 0.14$ | – |
| FT | $76.81 \pm 0.83$ | $94.59 \pm 0.08$ | $67.85 \pm 0.03$ | – | $78.02 \pm 18.48$ | $94.28 \pm 0.03$ | $67.39 \pm 0.03$ | – |
| NegGrad | $69.94 \pm 0.14$ | $70.43 \pm 1.52$ | $62.5 \pm 0.70$ | $0.053 \pm 0.003$ | $22.53 \pm 16.39$ | $74.81 \pm 2.09$ | $66.11 \pm 2.12$ | $0.187 \pm 0.129$ |
| NegGrad+ | $78.98 \pm 0.56$ | $79.23 \pm 0.37$ | $69.74 \pm 0.00$ | $0.066 \pm 0.004$ | $28.92 \pm 21.62$ | $75.94 \pm 1.72$ | $67.1 \pm 1.78$ | $0.226 \pm 0.177$ |
| FISHER | $78.85 \pm 0.70$ | $78.96 \pm 0.44$ | $69.59 \pm 0.09$ | $0.066 \pm 0.005$ | $91.64 \pm 4.87$ | $79.03 \pm 0.00$ | $69.64 \pm 0.00$ | $0.657 \pm 0.034$ |
| Influence | $78.98 \pm 0.68$ | $79.22 \pm 0.45$ | $69.74 \pm 0.02$ | $0.066 \pm 0.004$ | $78.82 \pm 18.09$ | $78.90 \pm 0.17$ | $69.52 \pm 0.12$ | $0.566 \pm 0.128$ |
| SCRUB | $78.86 \pm 0.70$ | $84.03 \pm 0.48$ | $69.47 \pm 0.11$ | $0.067 \pm 0.006$ | $0.53 \pm 0.55$ | $83.61 \pm 0.01$ | $69.42 \pm 0.09$ | $\underline{0.006 \pm 0.004}$ |
| DUCK | $67.22 \pm 1.29$ | $99.88 \pm 0.00$ | $61.87 \pm 0.01$ | $0.074 \pm 0.011$ | $0.00 \pm 0.00$ | $99.94 \pm 0.00$ | $61.37 \pm 0.74$ | $0.060 \pm 0.006$ |
| GDR-GMA | $67.35 \pm 1.32$ | $99.97 \pm 0.02$ | $66.98 \pm 0.38$ | $\underline{0.037 \pm 0.011}$ | $0.74 \pm 0.29$ | $99.22 \pm 0.22$ | $64.74 \pm 0.38$ | $0.043 \pm 0.000$ |
| SSD | $76.28 \pm 1.61$ | $76.17 \pm 0.85$ | $67.14 \pm 0.90$ | $0.065 \pm 0.005$ | $0.00 \pm 0.00$ | $78.65 \pm 0.31$ | $69.14 \pm 0.23$ | $\mathbf{0.004 \pm 0.001}$ |
| SCAR | $64.41 \pm 8.89$ | $75.51 \pm 3.52$ | $61.20 \pm 6.33$ | $0.100 \pm 0.109$ | $0.00 \pm 0.00$ | $76.47 \pm 0.40$ | $65.17 \pm 0.65$ | $0.033 \pm 0.005$ |
| OrthoGrad | $69.95 \pm 0.15$ | $82.27 \pm 0.87$ | $67.59 \pm 0.48$ | $\mathbf{0.016 \pm 0.002}$ | $0.48 \pm 0.40$ | $77.24 \pm 1.22$ | $67.25 \pm 1.24$ | $0.021 \pm 0.005$ |

The results are presented in Tables 4, 5. These experiments show that our method consistently meets the unlearning target and achieves superior performance in most settings, particularly under random

sampling. OrthoGrad also generalizes effectively across different scenarios, performing competitively in class-forgetting tasks. In contrast, several baselines, such as SSD, SCRUB, and SCAR, lack consistency, performing well in one setup but poorly in another. Additionally, SCAR and DUCK are designed specifically for image classification (see Section 5.1), and SCAR is only practical when a moderate amount of retain data is available (see Section 5.3.1). In conclusion, OrthoGrad delivers the most robust performance in both unlearning setups while being task-agnostic.

Table 5: *ImageNet with ViT architecture.* Performance is measured under two unlearning scenarios: random sampling of training data (3-seed average) and class removal (3-class average).

| | Random Sampling | | | | Class Forgetting | | | |
|---|---|---|---|---|---|---|---|---|
| Method | $\mathcal{A}_u$ | $\mathcal{A}_r$ | $\mathcal{A}_{test}$ | UIS ($\downarrow$) | $\mathcal{A}_u$ | $\mathcal{A}_r$ | $\mathcal{A}_{test}$ | UIS ($\downarrow$) |
| Original | $94.26 \pm 0.19$ | $94.04 \pm 0.24$ | $81.06 \pm 0.00$ | – | $98.1 \pm 1.62$ | $94.17 \pm 0.00$ | $81.06 \pm 0.00$ | – |
| Retrain | $3.36 \pm 0.04$ | $98.72 \pm 0.06$ | $3.42 \pm 0.04$ | – | $0.00 \pm 0.00$ | $95.08 \pm 0.15$ | $3.29 \pm 0.16$ | – |
| FT | $89.96 \pm 0.56$ | $99.94 \pm 0.01$ | $75.03 \pm 0.30$ | – | $91.97 \pm 8.18$ | $99.92 \pm 0.02$ | $74.07 \pm 0.05$ | – |
| NegGrad | $24.2 \pm 13.23$ | $24.26 \pm 13.46$ | $21.06 \pm 14.06$ | $0.720 \pm 0.186$ | $0.41 \pm 0.34$ | $89.63 \pm 3.84$ | $76.53 \pm 4.16$ | $0.030 \pm 0.028$ |
| NegGrad+ | $78.75 \pm 3.09$ | $87.63 \pm 5.34$ | $72.16 \pm 4.53$ | $0.069 \pm 0.051$ | $0.05 \pm 0.07$ | $90.67 \pm 4.22$ | $77.17 \pm 4.73$ | $0.024 \pm 0.028$ |
| FISHER | $94.36 \pm 0.09$ | $94.01 \pm 0.19$ | $81.00 \pm 0.05$ | $0.082 \pm 0.000$ | $98.02 \pm 1.82$ | $94.14 \pm 0.00$ | $80.99 \pm 0.00$ | $0.605 \pm 0.011$ |
| Influence | $89.08 \pm 1.90$ | $91.61 \pm 1.78$ | $77.48 \pm 1.68$ | $0.071 \pm 0.001$ | $15.28 \pm 20.76$ | $91.54 \pm 2.21$ | $78.13 \pm 2.06$ | $0.112 \pm 0.121$ |
| SCRUB | $94.54 \pm 0.15$ | $96.00 \pm 0.11$ | $80.76 \pm 0.05$ | $0.084 \pm 0.001$ | $45.82 \pm 37.8$ | $96.18 \pm 2.91$ | $75.81 \pm 4.13$ | $0.314 \pm 0.286$ |
| DUCK | $76.75 \pm 0.47$ | $100.00 \pm 0.00$ | $69.45 \pm 0.49$ | $0.097 \pm 0.000$ | $0.00 \pm 0.00$ | $99.39 \pm 0.02$ | $70.68 \pm 0.29$ | $0.064 \pm 0.001$ |
| GDR-GMA | $80.41 \pm 0.34$ | $99.34 \pm 0.05$ | $75.47 \pm 0.16$ | $\underline{0.038 \pm 0.003}$ | $0.15 \pm 0.16$ | $97.56 \pm 0.18$ | $77.03 \pm 0.28$ | $0.025 \pm 0.001$ |
| SSD | $93.5 \pm 0.19$ | $93.38 \pm 0.29$ | $80.35 \pm 0.17$ | $0.081 \pm 0.000$ | $0.00 \pm 0.00$ | $94.22 \pm 0.03$ | $80.9 \pm 0.03$ | $\mathbf{0.001 \pm 0.000}$ |
| SCAR | $71.49 \pm 1.34$ | $93.53 \pm 0.17$ | $77.83 \pm 0.15$ | $0.079 \pm 0.009$ | $0.00 \pm 0.00$ | $93.26 \pm 0.14$ | $76.85 \pm 0.24$ | $0.026 \pm 0.001$ |
| OrthoGrad | $81.04 \pm 0.13$ | $97.59 \pm 0.11$ | $78.22 \pm 0.13$ | $\mathbf{0.018 \pm 0.001}$ | $0.10 \pm 0.14$ | $94.26 \pm 0.04$ | $80.73 \pm 0.05$ | $\underline{0.002 \pm 0.000}$ |

### 5.3.1 ROBUSTNESS TO RETAIN SIZE

Here, we assess the robustness of our method to variations in the retain set size. To do so, we revisit the random sampling image classification setup from Section 5.3. Specifically, we experiment with varying retain dataset sizes, ranging from 1K to 200K samples, reporting the UIS values. The results are presented in Figure 2. We note that NegGrad is not affected by the size of the retain set since it only performs gradient ascent in the direction of the unlearn set. Additionally, we exclude SCAR from this experiment, as it involves inverting a covariance matrix, which results in a non-invertible matrix in the extreme case of 1K samples, and leads to memory overflow for 150K and 200K samples. Our model consistently outperforms baseline methods across all retain set sizes.

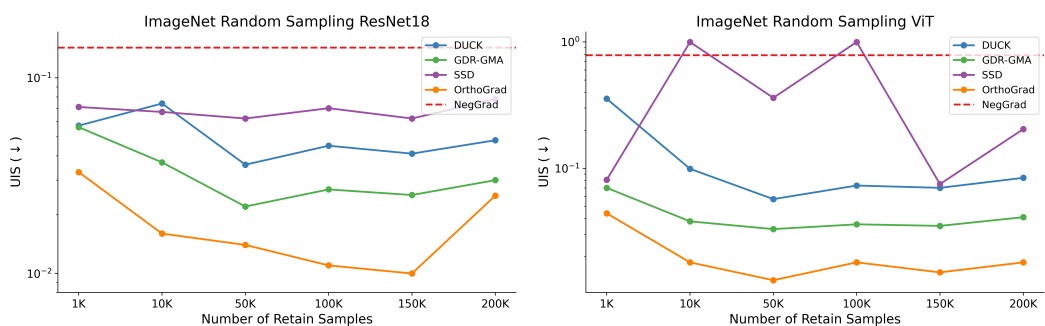

Figure 2: *Varying retain samples.* We report UIS values across varying numbers of retained samples.

## 6 CONCLUSIONS

In this work, we focus on machine unlearning in the low data regime, where access to the retain data is limited. We present OrthoGrad, a novel machine unlearning method that projects the aggregated unlearn gradient onto the subspace orthogonal to the individual gradients of the retain batch. We demonstrated the benefit of using per-sample gradient in the retain batch instead of averaging the retain gradients. Then, we demonstrate through various datasets, architectures, and unlearning setups the superiority of OrthoGrad over existing machine unlearning methods. Since OrthoGrad works well even without access to a large retain set, it can be applied in real-life use-cases where training data availability is constrained.

## REPRODUCIBILITY STATEMENT

Upon acceptance, we will release our code under a permissive license, including all training and evaluation scripts. Implementation details sufficient to reproduce our results are provided in Appendix B.

## ETHICS STATEMENT.

We emphasize that our approach does not introduce additional risks to individuals, as experiments are conducted exclusively on publicly available benchmark datasets. We do not process any personally identifiable or sensitive information. By facilitating efficient unlearning, this work may strengthen user rights and promote greater accountability in machine learning systems.

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

---

**Algorithm 1** *OrthoGrad*

---

**Input:** Forget set $\mathcal{D}_u$, retain set $\mathcal{D}_r$, learning rate $\eta$, combination parameter $\alpha$
**Output:** Updated model parameters $\theta_p$
Apply LoRA modules to the pretrained model:
$\theta_l = LoRA(\theta_p)$
**repeat**
    Sample a batch $\mathcal{B}_u \subset \mathcal{D}_u$ and $\mathcal{B}_r \subset \mathcal{D}_r$
    Compute the gradient $g_u$ from $\mathcal{B}_u$
    Compute the retain batch per-sample gradient matrix:
    $G_r = [g_r^1, g_r^2, \ldots, g_r^k]$ from $\mathcal{B}_r$
    Perform QR decomposition on $G_r$ to extract subspace:
    $Q = QR(G_r), \quad Q = [q_r^1, q_r^2, \ldots, q_r^k]$
    Project $g_u$ onto the retain gradient subspace:
    $p_i = \langle g_u, q_r^i \rangle q_r^i$
    Compute the orthogonalized unlearn gradient:
    $g_u^\perp = g_u - \sum_{i=1}^k p_i$
    Compute the mean retain gradient:
    $\bar{g}_r = \frac{1}{k} \sum_{i=1}^k g_r^i$
    Combine gradients to form a unified update direction:
    $g = \alpha \bar{g}_r - (1-\alpha) g_u^\perp$
    Update model parameters:
    $\theta_l \leftarrow \theta_l - \eta g$
**until** Convergence or maximum number of iterations
Merge LoRA modules:
$\theta_p = Merge(\theta_p, \theta_l)$

---

# A ADDITIONAL RESULTS

## A.1 IMAGE CLASSIFICATION

We revisit Section 5.3 and evaluate OrthoGrad on CIFAR-10 with a ResNet-18 backbone. In the random-sampling setting, we draw 5,000 images as the retain set and define the unlearn set as another 5,000 images sampled uniformly from the training data. In the class-forgetting setting, the unlearn set contains all training images from the designated unlearn class. In both cases, evaluation is performed on the standard test set. Results in Table 6 mirror our earlier findings: OrthoGrad reliably attains the unlearning objective, achieving superior performance in the class-forgetting setting while remaining comparable under random sampling.

Table 6: *CIFAR10 with ResNet18.* Performance is evaluated across two unlearning scenarios: random sampling (3-seed average) and class forgetting (3-class average).

| Method | Random Sampling | | | | Class Forgetting | | | |
|---|---|---|---|---|---|---|---|---|
| | $\mathcal{A}_u$ | $\mathcal{A}_r$ | $\mathcal{A}_{test}$ | UIS ($\downarrow$) | $\mathcal{A}_u$ | $\mathcal{A}_r$ | $\mathcal{A}_{test}$ | UIS ($\downarrow$) |
| Original | $96.1 \pm 0.28$ | $96.38 \pm 0.33$ | $81.97 \pm 0.00$ | $-$ | $97.3 \pm 1.21$ | $95.91 \pm 0.22$ | $81.97 \pm -$ | $-$ |
| Retrain | $60.43 \pm 0.22$ | $100.00 \pm 0.00$ | $61.04 \pm 0.35$ | $-$ | $0.00 \pm 0.00$ | $100.00 \pm 0.00$ | $55.55 \pm 1.03$ | $-$ |
| FT | $85.94 \pm 6.34$ | $88.49 \pm 7.44$ | $71.52 \pm 6.61$ | $-$ | $93.79 \pm 2.36$ | $100.00 \pm 0.00$ | $82.86 \pm 0.18$ | $-$ |
| NegGrad | $40.39 \pm 22.59$ | $39.78 \pm 23.17$ | $34.94 \pm 22.87$ | $0.540 \pm 0.308$ | $0.00 \pm 0.00$ | $18.72 \pm 10.03$ | $15.43 \pm 9.40$ | $0.405 \pm 0.057$ |
| NegGrad+ | $81.42 \pm 0.63$ | $83.00 \pm 0.59$ | $69.25 \pm 1.08$ | $0.082 \pm 0.009$ | $24.54 \pm 33.85$ | $70.56 \pm 25.45$ | $56.42 \pm 24.31$ | $0.305 \pm 0.164$ |
| FISHER | $72.29 \pm 6.59$ | $72.57 \pm 6.55$ | $61.60 \pm 6.75$ | $0.183 \pm 0.089$ | $84.98 \pm 21.79$ | $69.08 \pm 5.77$ | $61.33 \pm 1.92$ | $0.645 \pm 0.143$ |
| Influence | $11.55 \pm 1.42$ | $11.59 \pm 1.30$ | $11.31 \pm 1.48$ | $0.860 \pm 0.019$ | $43.07 \pm 29.16$ | $66.72 \pm 31.99$ | $54.53 \pm 25.95$ | $0.431 \pm 0.043$ |
| SCRUB | $40.18 \pm 5.37$ | $42.51 \pm 4.88$ | $38.58 \pm 4.30$ | $0.519 \pm 0.066$ | $1.75 \pm 1.61$ | $87.32 \pm 3.09$ | $64.7 \pm 1.70$ | $0.116 \pm 0.022$ |
| DUCK | $86.46 \pm 0.19$ | $99.5 \pm 0.29$ | $78.02 \pm 0.35$ | $\underline{0.051 \pm 0.002}$ | $0.00 \pm 0.00$ | $41.28 \pm 4.25$ | $35.35 \pm 4.82$ | $0.284 \pm 0.029$ |
| GDR-GMA | $81.75 \pm 0.40$ | $99.08 \pm 0.41$ | $71.6 \pm 0.28$ | $0.065 \pm 0.002$ | $0.19 \pm 0.19$ | $89.88 \pm 5.31$ | $66.79 \pm 3.30$ | $0.093 \pm 0.021$ |
| SSD | $96.10 \pm 0.23$ | $96.38 \pm 0.27$ | $81.97 \pm 0.00$ | $0.090 \pm 0.000$ | $0.040 \pm 0.06$ | $80.49 \pm 2.60$ | $61.99 \pm 1.41$ | $0.122 \pm 0.008$ |
| SCAR | $81.38 \pm 1.15$ | $96.07 \pm 0.17$ | $78.92 \pm 0.84$ | $\mathbf{0.024 \pm 0.010}$ | $0.00 \pm 0.00$ | $91.39 \pm 3.09$ | $72.48 \pm 3.41$ | $\underline{0.058 \pm 0.021}$ |
| OrthoGrad | $81.18 \pm 2.92$ | $93.27 \pm 0.71$ | $73.35 \pm 0.41$ | $0.058 \pm 0.005$ | $0.36 \pm 0.35$ | $97.57 \pm 0.43$ | $74.87 \pm 1.05$ | $\mathbf{0.045 \pm 0.006}$ |

## A.2 ABLATION STUDY - IMAGE CLASSIFICATION

In this section, we analyze the individual contributions of each component in OrthoGrad, following a similar procedure to Section 5.1.1. Specifically, we revisit the image classification unlearning

setup described in Section 5.3 and compare the following variants: OrthoGrad Mean, OrthoGrad Per-sample, OrthoGrad Mean/Per-sample+LoRA, and the full OrthoGrad approach. The experiments are conducted using a ResNet18 model trained on the CIFAR10 dataset for the unlearning task. The results, averaged over 3 seeds, are presented in Table 7. Notably, OrthoGrad achieves higher test accuracy and lower UIS values, indicating superior unlearning effectiveness without sacrificing generalization. This highlights the importance of combining both per-sample gradient components and the low-ranking optimization strategy.

Table 7: *Image classification ablation study.* Evaluation of OrthoGrad variants on CIFAR10 random unlearning. Values are averaged over 3 different seeds.

| Method | $\mathcal{A}_u$ | $\mathcal{A}_r$ | $\mathcal{A}_{test}$ | UIS ($\downarrow$) |
|---|---|---|---|---|
| OrthoGrad mean | $82.47 \pm 0.51$ | $84.33 \pm 0.45$ | $70.06 \pm 0.19$ | $0.076 \pm 0.002$ |
| OrthoGrad Per sample | $82.10 \pm 0.21$ | $83.21 \pm 0.45$ | $71.75 \pm 0.42$ | $0.064 \pm 0.002$ |
| OrthoGrad Mean + LORA | $82.07 \pm 0.25$ | $84.83 \pm 1.03$ | $71.42 \pm 0.72$ | $0.066 \pm 0.004$ |
| OrthoGrad Per sample + LORA | $82.01 \pm 0.26$ | $82.86 \pm 0.45$ | $72.04 \pm 0.27$ | $0.062 \pm 0.002$ |
| **OrthoGrad** | $\mathbf{81.35 \pm 1.2}$ | $\mathbf{87.44 \pm 1.27}$ | $\mathbf{73.34 \pm 1.32}$ | $\mathbf{0.058 \pm 0.015}$ |

### A.3 THE RELATION BETWEEN RETAIN AND UNLEARN GRADIENTS

Here, we explore the relationship between the retain and unlearn gradients, which plays a central role in the effectiveness of our method. A natural concern is that if the gradients of the unlearn and retain sets are highly aligned, the orthogonal component of the unlearn gradient used in our projection step could be small. This may potentially weaken the unlearning effect. To address this concern, we analyze the cosine similarity between the retain and unlearn gradients over the course of training. Specifically, we repeat the class and random forgetting experiments on the ImageNet dataset using the ResNet18 architecture and report the cosine similarity between the unlearn and retain gradients. The results are presented in Figure 3. The results show that, for the majority of the unlearning process, the gradients are not highly aligned—indicated by consistently non-zero cosine similarity values. These observations highlight the importance of the projection step in OrthoGrad.

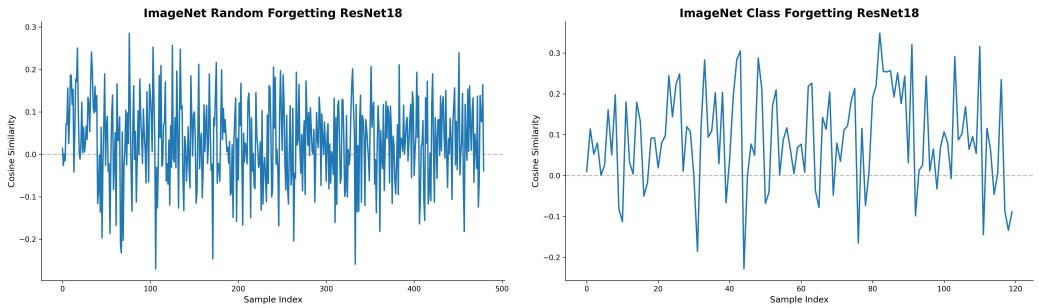

Figure 3: Cosine similarity between retain and unlearn gradients during the unlearning process on ImageNet using ResNet18. The non-zero similarity values throughout training indicate that the gradients are not highly aligned, validating the importance of the projection step in OrthoGrad.

### A.4 ROBUSTNESS TO $\alpha$

In Section 4.2, we describe the unlearning update direction defined by OrthoGrad. This update rule incorporates a balancing parameter $\alpha$, which interpolates between the unlearn and retain gradients. Here, we conduct an ablation study to assess the robustness of our method w.r.t $\alpha$. Specifically, we run the class forgetting experiment on Imagenet using the ViT-b16 architecture across varying values $\alpha \in \{0, 0.3, 0.7, 0.9\}$. The results are presented in Figure 4. Notably, similar to the approach in Bonato et al. (2024), OrthoGrad is also capable of operating solely based on the unlearn gradient by setting $\alpha=0$. In this case, the update reduces to performing gradient ascent without relying on the retain set.

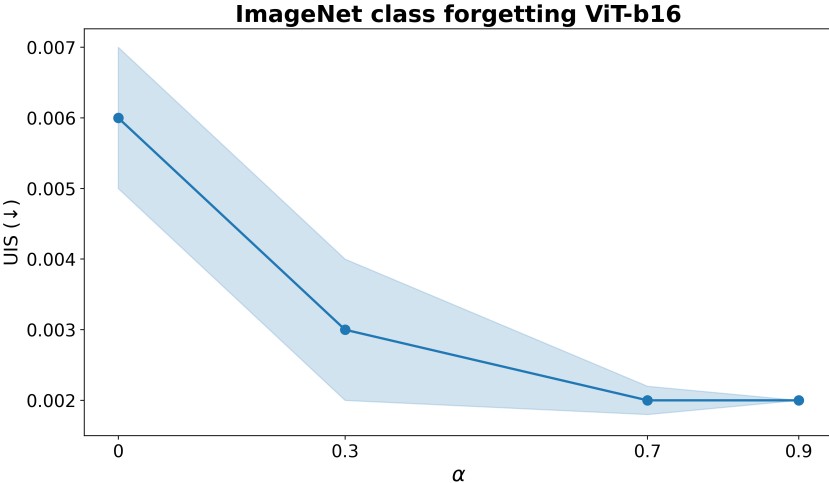

Figure 4: Comparison across different values of $\alpha$ for OrthoGrad on the ImageNet class forgetting setup using ViT.

### A.5 MIA AS MACHINE UNLEARNING METRIC

MIA can serve as a useful metric for evaluating machine unlearning. However, MIA has fundamental limitations that make it insufficient as a standalone measure of unlearning quality. Notably, an MIA score of 0, often interpreted as a perfect result, can indicate that the model has undergone catastrophic forgetting. In such cases, the model may not only forget the targeted data but also lose generalization capabilities. This shortcoming has been highlighted in prior work. Foster et al. (2024) point out that low MIA accuracy may result from aggressive dampening that leads to degraded model performance. Similarly, Hayes et al. (2024) emphasized that relying solely on MIA can produce a misleading sense of unlearning efficacy, particularly when models are overly perturbed or when forgetting extends beyond the target data. Given these limitations, we believe MIA should be considered alongside other metrics that directly measure the trade-off between forgetting and retention. For completeness, we revisited the class forgetting benchmark on ImageNet using the ViT architecture and report the corresponding MIA scores. The results are presented in Table 8.

Table 8: *ImageNet using ViT-b16*. The results on class sampling setup are averaged over 3 classes.

| Method | $\mathcal{A}_u$ | $\mathcal{A}_r$ | $\mathcal{A}_{test}$ | UIS ($\downarrow$) | MIA ($\downarrow$) |
|---|---|---|---|---|---|
| Original | $94.26 \pm 0.19$ | $94.04 \pm 0.24$ | $81.06 \pm 0.00$ | $-$ | $-$ |
| Retrain | $0.00 \pm 0.00$ | $95.08 \pm 0.15$ | $3.29 \pm 0.16$ | $-$ | $-$ |
| FT | $91.97 \pm 8.18$ | $99.92 \pm 0.02$ | $74.07 \pm 0.05$ | $-$ | $-$ |
| NegGrad | $0.41 \pm 0.34$ | $89.63 \pm 3.84$ | $76.53 \pm 4.16$ | $0.030 \pm 0.028$ | $0.23 \pm 0.39$ |
| NegGrad+ | $0.05 \pm 0.07$ | $90.67 \pm 4.22$ | $77.17 \pm 4.73$ | $0.024 \pm 0.028$ | $0.38 \pm 0.12$ |
| FISHER | $98.02 \pm 1.82$ | $94.14 \pm 0.00$ | $80.99 \pm 0.00$ | $0.605 \pm 0.011$ | $0.83 \pm 0.00$ |
| Influence | $15.28 \pm 20.76$ | $91.54 \pm 2.21$ | $78.13 \pm 2.06$ | $0.112 \pm 0.121$ | $0.04 \pm 0.06$ |
| SCRUB | $45.82 \pm 37.80$ | $96.18 \pm 2.91$ | $75.81 \pm 4.13$ | $0.314 \pm 0.286$ | $0.84 \pm 0.01$ |
| DUCK | $0.00 \pm 0.00$ | $99.39 \pm 0.02$ | $70.68 \pm 0.29$ | $0.064 \pm 0.001$ | $0.11 \pm 0.17$ |
| GDR-GMA | $0.15 \pm 0.16$ | $97.56 \pm 0.18$ | $77.03 \pm 0.28$ | $0.025 \pm 0.001$ | $0.54 \pm 0.41$ |
| OrthoGrad | $0.10 \pm 0.14$ | $94.26 \pm 0.04$ | $80.73 \pm 0.05$ | $0.002 \pm 0.000$ | $0.30 \pm 0.24$ |

### A.6 BENEFITS OF INTEGRATING LoRA INTO ORTHOGRAD

In Section 4.2, we detail the steps of OrthoGrad, including the integration of LoRA modules. Here, we further elaborate on the motivation behind this design choice and highlight the specific benefits

LoRA brings to scalable machine unlearning. One key advantage of using LoRA is its ability to localize parameter updates, which intuitively helps reduce unintended interference with retained knowledge. By limiting the impact of unlearning to a low-rank subspace, LoRA allows for more controlled and precise modifications, which aligns well with the goal of minimizing collateral forgetting. Additionally, LoRA is significantly more parameter-efficient than fine-tuning full model weights. This not only leads to reduced memory usage but also lowers the computational cost, making the approach viable for large-scale models like Whisper. These advantages collectively make LoRA a natural fit for OrthoGrad, enabling both effective unlearning and practical deployment in real-world systems.

## A.7 STANDARD MACHINE UNLEARNING SETUP

This work addresses the low-data regime, where the available retain dataset is limited in size. We evaluate OrthoGrad within the standard machine unlearning framework, in which the retain set coincides with the original training set used for the pretrained model. The experiments are conducted on the CIFAR10 dataset using the ResNet18 architecture, considering both random sampling and class forgetting scenarios. The results are presented in Table 9. We show that OrthoGrad remains effective even when the retain set is relatively large, as further discussed in Section 5.3.1.

Table 9: *CIFAR10 with ResNet18.* Performance is evaluated across two unlearning scenarios: random sampling (3-seed average) and class forgetting (3-class average).

| Method | Random Sampling | | | | Class Forgetting | | | |
|---|---|---|---|---|---|---|---|---|
| | $\mathcal{A}_u$ | $\mathcal{A}_r$ | $\mathcal{A}_{test}$ | UIS ($\downarrow$) | $\mathcal{A}_u$ | $\mathcal{A}_r$ | $\mathcal{A}_{test}$ | UIS ($\downarrow$) |
| Original | $96.38 \pm 0.37$ | $96.09 \pm 0.04$ | $81.97 \pm 0.00$ | $-$ | $97.3 \pm 1.21$ | $95.98 \pm 0.13$ | $81.97 \pm 0.00$ | $-$ |
| Retrain | $82.09 \pm 0.3$ | $99.9 \pm 0.1$ | $82.6 \pm 0.63$ | $-$ | $0.00 \pm 0.00$ | $99.67 \pm 0.31$ | $74.22 \pm 1.56$ | $-$ |
| NegGrad | $77.47 \pm 2.11$ | $77.84 \pm 1.76$ | $64.43 \pm 1.4$ | $0.134 \pm 0.024$ | $0.00 \pm 0.00$ | $19.23 \pm 11.41$ | $16.17 \pm 10.59$ | $0.401 \pm 0.064$ |
| NegGrad+ | $96.28 \pm 0.07$ | $96.05 \pm 0.34$ | $81.98 \pm 0.2$ | $0.088 \pm 0.002$ | $18.69 \pm 25.85$ | $68.64 \pm 25.74$ | $54.26 \pm 24.24$ | $0.283 \pm 0.120$ |
| FISHER | $96.3 \pm 0.48$ | $96.00 \pm 0.04$ | $81.83 \pm 0.1$ | $0.088 \pm 0.002$ | $0.06 \pm 0.05$ | $11.54 \pm 0.36$ | $10.37 \pm 0.265$ | $0.437 \pm 0.001$ |
| Influence | $96.31 \pm 0.4$ | $96.07 \pm 0.04$ | $81.96 \pm 0.05$ | $0.087 \pm 0.002$ | $17.54 \pm 12.92$ | $63.74 \pm 23.27$ | $50.92 \pm 17.18$ | $0.296 \pm 0.084$ |
| SCRUB | $55.58 \pm 2.26$ | $56.73 \pm 2.37$ | $55.23 \pm 3.14$ | $0.324 \pm 0.036$ | $0.02 \pm 0.00$ | $94.02 \pm 2.3$ | $72.91 \pm 0.76$ | $\underline{0.055} \pm \underline{0.004}$ |
| DUCK | $81.71 \pm 1.43$ | $89.34 \pm 2.06$ | $82.03 \pm 1.40$ | $\mathbf{0.015} \pm \mathbf{0.009}$ | $0.00 \pm 0.00$ | $79.12 \pm 13.5$ | $66.93 \pm 9.75$ | $0.091 \pm 0.072$ |
| GDR-GMA | $81.1 \pm 0.70$ | $86.18 \pm 3.26$ | $73.16 \pm 2.59$ | $0.059 \pm 0.020$ | $0.00 \pm 0.00$ | $87.99 \pm 4.33$ | $67.94 \pm 5.25$ | $0.085 \pm 0.032$ |
| SSD | $25.35 \pm 39.73$ | $94.95 \pm 1.22$ | $74.23 \pm 1.77$ | $0.200 \pm 0.230$ | $25.35 \pm 32.44$ | $94.95 \pm 1.00$ | $74.23 \pm 1.44$ | $0.202 \pm 0.189$ |
| SCAR | $80.11 \pm 1.65$ | $91.06 \pm 1.10$ | $79.21 \pm 1.28$ | $\underline{0.029} \pm \underline{0.016}$ | $0.00 \pm 0.00$ | $87.71 \pm 0.85$ | $70.75 \pm 1.32$ | $0.068 \pm 0.008$ |
| OrthoGrad | $81.35 \pm 1.2$ | $87.44 \pm 1.27$ | $73.34 \pm 1.32$ | $0.058 \pm 0.015$ | $0.67 \pm 0.29$ | $96.9 \pm 0.32$ | $75.67 \pm 0.93$ | $\mathbf{0.042} \pm \mathbf{0.007}$ |

## A.8 EFFECTIVENESS OF ORTHOGRAD IN THE PRESENCE OF LARGER FORGET SETS

To evaluate OrthoGrad on larger forget sets, we extend the class forgetting setup to simultaneously remove three classes. Using the ResNet-18 architecture on CIFAR-10, we repeat the experiment across three different class combinations and report the mean and standard deviation of the results (Table 10). The retain set consists of 5K images, consistent with the original setup.

Table 10: *Image classification unlearning results.* Comparison of OrthoGrad and GDR-GMA on CIFAR10 class unlearning. Values are averaged over 3 different combinations of classes.

| Method | $\mathcal{A}_u$ | $\mathcal{A}_r$ | $\mathcal{A}_{test}$ | $\mathcal{A}_{test}$ (w/o unlearned classes) | UIS ($\downarrow$) | UIS (w/o unlearned classes) ($\downarrow$) |
|---|---|---|---|---|---|---|
| GDR-GMA | $0.48 \pm 0.04$ | $95.80 \pm 2.58$ | $56.94 \pm 0.41$ | $81.20 \pm 0.57$ | $0.155 \pm 0.002$ | $0.024 \pm 0.014$ |
| **OrthoGrad** | $\mathbf{0.0 \pm 0.0}$ | $\mathbf{97.15 \pm 1.13}$ | $\mathbf{59.11 \pm 2.82}$ | $\mathbf{84.43 \pm 4.21}$ | $\mathbf{0.139 \pm 0.014}$ | $\mathbf{0.014 \pm 0.004}$ |

The overall accuracy is naturally lower because of the removal of three classes. Therefore, we also report the accuracy of the original test set after removing the samples of the unlearned classes. Nevertheless, OrthoGrad achieves better unlearning performance and higher test accuracy compared to GDR-GMA.

Additionally, we conduct random forgetting experiments on ImageNet with the ResNet-18 architecture. The unlearn sets contain 50K, 150K, and 200K samples, corresponding to 10×, 30×, and 40× the size of the unlearn set in our main experiments. The retain set is fixed at 10K samples. The results are shown in Table 11.

Table 11: *Random unlearning on ImageNet with varying unlearn set sizes.* Evaluation of OrthoGrad vs. GDR-GMA on ImageNet with 50K, 150K, and 200K unlearn sets. Values shown are averaged over 3 seeds.

| Method | 50K samples | | | | 150K samples | | | | 200K samples | | | |
|---|---|---|---|---|---|---|---|---|---|---|---|---|
| | $\mathcal{A}_u$ | $\mathcal{A}_r$ | $\mathcal{A}_{test}$ | UIS | $\mathcal{A}_u$ | $\mathcal{A}_r$ | $\mathcal{A}_{test}$ | UIS | $\mathcal{A}_u$ | $\mathcal{A}_r$ | $\mathcal{A}_{test}$ | UIS |
| Original | 79.26 | 79.17 | 69.76 | - | 79.32 | 79.17 | 69.76 | - | 79.31 | 79.17 | 69.76 | - |
| GDR-GMA | 68.43 | 100.0 | 63.33 | 0.05566 | 68.85 | 100.0 | 62.13 | 0.06123 | 68.60 | 100.0 | 61.75 | 0.06573 |
| OrthoGrad | 70.09 | 89.41 | 63.53 | **0.04700** | 69.48 | 91.45 | 62.21 | **0.05608** | 69.75 | 91.11 | 62.35 | **0.05317** |

These results show that OrthoGrad remains effective with larger unlearn sets in the random forgetting setup. Compared to GDR-GMA, it achieves unlearn accuracy closer to the original model's test accuracy, leading to better overall unlearning process.

## A.9 BATCH COMPOSITION

Here, we conduct an ablation study to examine the effect of retain set batch composition. Specifically, we compare the random batch sampling used in our main experiments with an alternative strategy in which each retain batch is restricted to a single class. To this end, we revisit the CIFAR10 class forgetting experiment and evaluate OrthoGrad under both settings: (i) the original random sampling and (ii) single-class retain batches (excluding the forget class). Results are reported in Table 12.

Table 12: *Ablation on batch composition.* Performance of OrthoGrad with and without sampling from same-class batches.

| Method | $\mathcal{A}_u$ | $\mathcal{A}_r$ | $\mathcal{A}_{\text{test}}$ | UIS ($\downarrow$) |
|---|---|---|---|---|
| OrthoGrad same class batch | $0.28 \pm 0.20$ | $94.28 \pm 1.26$ | $72.91 \pm 0.04$ | $0.057 \pm 0.001$ |
| OrthoGrad | $0.36 \pm 0.35$ | $97.57 \pm 0.43$ | $74.87 \pm 1.05$ | $0.045 \pm 0.006$ |

The results suggest that using shuffled retain batches leads to better performance for OrthoGrad. We hypothesize that this is because shuffled batches enable smoother optimization. In contrast, projecting the unlearn gradient onto the subspace spanned by a single class may constrain the optimization dynamics.

## A.10 PROJECTION-INDUCED SIGNAL LOSS

A potential concern with our approach is that by projecting the unlearn gradient away from the retain gradient directions, part of the unlearning signal could be lost. To study this, we measure the ratio between the projection component and the unlearn gradient, $\frac{\|g_u^\perp\|}{\|g_u\|}$, where $g_u$ is the gradient from the forget set and $g_u^\perp$ is its projection component. This ratio indicates how much of the unlearning signal is preserved after projection.

We conduct experiments on CIFAR-10 with ResNet-18 under two setups: class forgetting and random forgetting. In each case, we track the projection ratio reporting its mean, standard deviation, and range. The results indicate that while part of the signal is removed during projection, the ratio remains consistently above zero, showing that sufficient unlearning signal is retained to remain effective. Specifically, in the class forgetting setting, we obtain $0.241 \pm 0.036$ (min: 0.181, max: 0.597), and in the random forgetting setting $0.229 \pm 0.028$ (min: 0.176, max: 0.535). The evolution of this ratio over training steps is illustrated in Figure 5.

## B EXPERIMENTAL DETAILS

### B.1 IMAGE CLASSIFICATION

We provide additional details about the image classification machine unlearning setup, including general information and the hyperparameter search performed for each method. This setup leverages

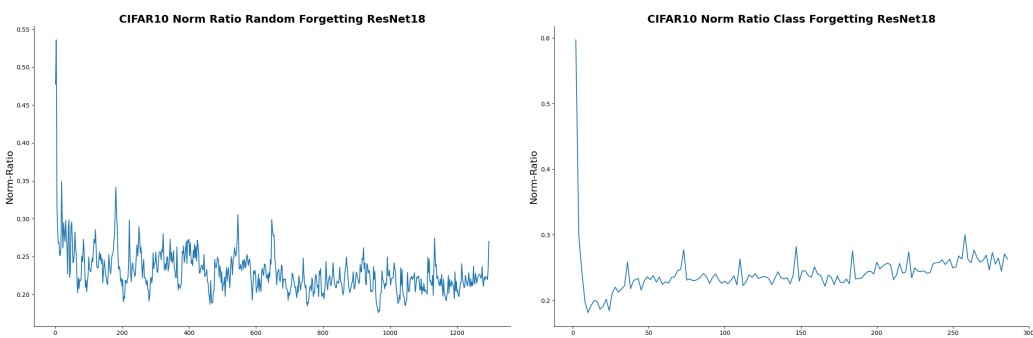

Figure 5: Measured ratio between the projection component and the unlearn gradient conducted on CIFAR-10 with ResNet-18 under two setups class forgetting and random forgetting.

the splits from the CIFAR10 and ImageNet datasets, focusing on either random unlearning samples or specific class samples. Results are reported across three random seeds or three different classes. All methods are trained for 30 epochs, with each setup utilizing its corresponding stopping criteria as explained in 5.3.

**Retrain.** We performed grid search for learning rate ($\eta$), and batch size for $\mathcal{D}_r$. Specifically, for $\eta$, we searched over $\{1e-3, 1e-2, 1e-1, 3e-2\}$, and for retain batch size, we searched over $\{256, 128\}$. **For ViT architecture** The optimal parameters for random and class forgetting respectively $\eta = (3e-2, 3e-2)$, retain batch size = $(128, 128)$. **For ResNet18 architecture on ImageNet** The optimal parameters for random and class forgetting respectively $\eta = (1e-1, 1e-1)$, retain batch size = $(128, 128)$. **For ResNet18 architecture on CIFAR-10** The optimal parameters for random and class forgetting respectively $\eta = (1e-2, 1e-2)$, retain batch size = $(128, 128)$. Training was performed for 100 epochs for ImageNet-based models and 30 epochs for CIFAR-10.

**OrthoGrad.** We performed grid search for the combination ($\alpha$) and learning rate ($\eta$), and batch sizes for $\mathcal{D}_u$ and $\mathcal{D}_r$. We apply LoRA modules to all linear layers within the self-attention and cross-attention layers. We set the rank to 8 and the scaling factor to 32. Specifically, for $\alpha$, we searched over $\{0.9, 0.8\}$, and for $\eta$, we searched over $\{0.001, 0.01\}$. For the retain batch size, we considered $\{256, 128\}$, and for the unlearn batch size, we searched over $\{256, 128\}$. **For ViT architecture** The optimal parameters for random and class forgetting respectively $\alpha = (0.9, 0.8)$, $\eta = (0.001, 0.001)$, retain batch size = $(128, 256)$, unlearn batch size = $(128, 256)$. **For ResNet18 architecture** The optimal parameters for random and class forgetting respectively $\alpha = (0.9, 0.8)$, $\eta = (0.001, 0.001)$, retain batch size = $(256, 256)$, unlearn batch size = $(128, 256)$.

**NegGrad.** We performed grid search for learning rate ($\eta$), and batch size for $\mathcal{D}_u$. Specifically, for $\eta$, we searched over $\{1e-3, 1e-4, 1e-5, 1e-6\}$, and for unlearn batch size, we searched over $\{256, 128\}$. **For ViT architecture** The optimal parameters for random and class forgetting respectively $\eta = (1e-4, 1e-4)$, unlearn batch size = $(256, 128)$. **For ResNet18 architecture** The optimal parameters for random and class forgetting respectively $\eta = (1e-5, 1e-5)$, unlearn batch size = $(128, 256)$.

**NegGrad+.** We performed grid search for learning rate ($\eta$), and batch sizes for $\mathcal{D}_u$ and $\mathcal{D}_r$. Specifically, for $\eta$, we searched over $\{1e-3, 1e-4, 1e-5, 1e-6\}$. For the retain batch size, we considered $\{256, 128\}$, and for unlearn batch size, we searched over $\{256, 128\}$. **For ViT architecture** The optimal parameters for random and class forgetting respectively $\eta = (1e-3, 1e-3)$, retain batch size = $(128, 256)$, unlearn batch size = $(128, 256)$. **For ResNet18 architecture** The optimal parameters for random and class forgetting respectively $\eta = (1e-6, 1e-5)$, retain batch size = $(256, 256)$, unlearn batch size = $(256, 256)$.

**Fisher.** We performed grid search for ($\alpha$), and batch size for $\mathcal{D}_r$. Specifically, for $\alpha$, we searched over $\{1e-7, 1e-8, 1e-9\}$, and for retain batch size, we searched over $\{128, 256\}$. **For ViT architecture** The optimal parameters for random and class forgetting respectively $\eta = (1e-9, 1e-9)$,

retain batch size = $(128, 256)$. **For ResNet18 architecture** The optimal parameters for random and class forgetting respectively $\eta = (1e-9, 1e-9)$, retain batch size = $(128, 256)$.

**Influence.** We performed grid search for $(\alpha)$, and batch sizes for $\mathcal{D}_u$ and $\mathcal{D}_r$. Specifically, for $\alpha$, we searched over $\{1, 0.1, 0.01, 1e-3\}$. For the retain batch size, we considered $\{64, 128, 256\}$, and for unlearn batch size, we searched over $\{64, 128, 256\}$. **For ViT architecture** The optimal parameters for random and class forgetting respectively $\eta = (1, 1)$, retain batch size = $(256, 128)$ and for unlearn batch size = $(128, 128)$. **For ResNet18 architecture** The optimal parameters for random and class forgetting respectively $\eta = (0.001, 1)$, retain batch size = $(64, 256)$ and for unlearn batch size = $(256, 128)$.

**SCRUB.** We performed grid search for learning rate $(\eta)$, and batch sizes for $\mathcal{D}_u$ and $\mathcal{D}_r$. Specifically, for $\eta$, we searched over $\{5e-2, 5e-5-3, 5e-4\}$. For the retain batch size, we considered $\{256, 512\}$, and for unlearn batch size, we searched over $\{256, 512\}$. **For ViT architecture** $\eta = 5e-3$ retain batch size = 256, unlearn batch size = 512. **For ResNet18 architecture** $\eta = 5e-4$ retain batch size = 256, unlearn batch size = 512.

**DUCK.** We performed grid search for learning rate $(\eta)$, and batch sizes for $\mathcal{D}_u$ and $\mathcal{D}_r$. Specifically, for $\eta$, we searched over $\{2e-4, 5e-4, 5e-5\}$. For the retain batch size, we considered $\{128, 1024\}$, and for unlearn batch size, we searched over $\{128, 1024\}$. **For ViT architecture** $\lambda_{fgt} = (1.5, 0.5)$, $\lambda_{ret} = (1.5, 1.5)$, batch ratio = $(5, 30)$, $\eta = (5e-5, 2e-4)$, retain and unlearn batch size = $(128, 128)$, $\tau = (3, 3)$. **For ResNet18 architecture** $\lambda_{fgt} = (1.5, 0.5)$, $\lambda_{ret} = (1.5, 1.5)$, batch ratio = $(5, 30)$, $\eta = (5e-4, 2e-4)$, retain and unlearn batch size = $(1024, 1024)$, temperature = $(3, 3)$.

**SCAR.** We performed grid search for learning rate $(\eta)$. Specifically, for $\eta$, we searched over $\{1e-4, 5e-4, 1e-3\}$. In addition, to ensure a fair comparison and maintain consistency with prior work, we adopted the remaining hyperparameters as reported in Bonato et al. (2024) (refer to Table 9 in Bonato et al. (2024) for details).

**SSD.** We performed grid search for $(\alpha)$, and batch sizes for $\mathcal{D}_u$ and $\mathcal{D}_r$. Specifically, for $\alpha$, we searched over $\{1.1, 1.3, 1.5, 1.7, 5, 10, 30, 50\}$. For the retain batch size, we considered $\{64, 128, 256\}$, and for unlearn batch size, we searched over $\{64, 128, 256\}$. **For ViT architecture** The optimal parameters for random and class forgetting respectively $\eta = (1.3, 30)$, retain batch size = $(128, 128)$ and for unlearn batch size = $(128, 256)$. **For ResNet18 architecture** The optimal parameters for random and class forgetting respectively $\eta = (10, 10)$, retain batch size = $(128, 128)$ and for unlearn batch size = $(256, 256)$.

**GDR-GMA.** We performed grid search for learning rate $(\eta)$, and batch sizes for $\mathcal{D}_u$ and $\mathcal{D}_r$. Specifically, for $\eta$, we searched over $\{1e-3, 1e-4, 1e-5\}$. For the retain batch size, we considered $\{128, 256\}$, and for unlearn batch size, we searched over $\{128, 256\}$. For all setups and architectures $\eta = 1e-4$ retain batch size = 256, unlearn batch size = 256.

## B.2 Automatic Speech Recognition

Here, we present additional details about the ASR machine unlearning setup. This includes both general information about the setup and the hyperparameter search conducted for each method. In this setup, we utilized the train-100 split from the LibriSpeech dataset, targeting unlearning samples from a single speaker. The results are reported across 5 randomly sampled speakers. For all methods, we utilize a batch size of 48 for the retain and unlearn sets and 30 epochs with early stopping. In addition, we use the well-known Adam Kingma (2014) as the optimizer.

**OrthoGrad.** - We performed grid search for the combination $(\alpha)$ and learning rate $(\eta)$ parameters. Specifically for $\alpha$ we searched over $\{0.05, 0.2, 0.35, 0.5\}$ and $\{1e-5, 5e-6, 1e-6\}$ for $\eta$. We apply LoRA modules to all linear layers within the self-attention and cross-attention layers. We set the rank to 8 and the scaling factor to 32.

**Finetune.** We performed a grid search for the learning rate parameter. Specifically, we explored learning rates in the range $\{1e-5, 5e-6, 1e-6\}$, and the optimal learning rate chosen is $\eta = 1e-5$.

**NegGrad+.** - We performed a grid search for the learning rate parameter. Specifically we searched over $\{1e-5, 5e-6, 2.5e-5, 1e-6, 5e-7, 1e-7\}$, and the optimal learning rate chosen is $5e-6$.

**SCRUB.** We performed a grid search for the learning rate and the number of epochs in which SCRUB performs unlearning steps. We searched over the range $[1e-4, 1e-7]$ with a step size of $0.5$ in multiplication, as well as $\{10, 20, 30\}$ for the number of unlearning epochs. We observed that SCRUB either failed to achieve unlearning entirely or caused the model to collapse, resulting in poor generalization. We report the results with $1e-5$ learning and $20$ unlearning epochs. We also used temperature rescaling of $4$ in the knowledge distillation loss and $5e-4$ weight decay.

**GDR-GMA.** We performed a grid search for the learning rate and learning rate scheduling factor parameters. We searched over the range $[1e-4, 1e-7]$ with a step size of $0.5$ in multiplication, and found the optimal learning rate to be $1e-4$. We explored learning rate scheduling factors in the set $\{10, 50, 100, 200\}$ and identified $100$ as the optimal value.

### B.3 DATA PREPROCESSING

We adopt the data preprocessing approach outlined by Radford et al. (2023). The audio samples are resampled to 16 kHz, and log-magnitude Mel-spectrograms are generated. Specifically, we compute 80-channel Mel-spectrograms using 25-millisecond windows with a 10-millisecond stride.

## C LIMITATIONS

This work focuses on the low-data regime, showing strong gains over baselines in this setting. However, per-sample gradient handling increases GPU memory use. Despite these limitations, OrthoGrad presents a compelling solution in data-constrained environments.

## D LLM USAGE

We used large language models to improve the readability of the manuscript, including grammar and clarity. All research ideas, experiments, and analyses were conducted and developed by the authors.

