# OpenReview forum: "Go Beyond Your Means: Unlearning with Per-Sample Gradient Orthogonalization"
_ICLR.cc/2026/Conference — ICLR 2026 Conference Withdrawn Submission_

### Official Review · Reviewer_BMtH · 2025-10-29

**Soundness:** 2
**Presentation:** 2
**Contribution:** 2
**Rating:** 4
**Confidence:** 3

**Summary:**

This paper proposes a machine unlearning method named OrthoGrad, designed for practical scenarios where access to the retain set is highly restricted. OrthoGrad leverages gradient orthogonalization to update the model parameters such that the unlearning gradient becomes orthogonal to the retain gradient, aiming to forget the unlearn set while preserving performance on the retain set. The authors validate OrthoGrad through experiments on both image classification and automatic speech recognition tasks, demonstrating its effectiveness.

**Strengths:**

- The paper considers a practical unlearning scenario where the retain set cannot be fully accessed, a relevant and underexplored problem in real-world applications. The authors provide insightful discussions and analyses of this setting.

- The paper identifies limitations in current evaluation metrics and introduces a new measure, the Unlearning Impact Score, offering a more nuanced evaluation of unlearning performance.

- The method is evaluated on diverse datasets and tasks, including both standard image classification and more practical automatic speech recognition, which helps validate its generality across modalities.

**Weaknesses:**

- Although OrthoGrad presents an interesting idea, the introduction does not sufficiently emphasize the method's novelty. Existing unlearning methods that do not require retain-set access (e.g., NegGrad, Boundary Shrink) are not adequately compared or discussed. Furthermore, OrthoGrad itself still depends on partial retain data, meaning it does not fully solve the "retain-restricted" problem.

- Since OrthoGrad continuously computes gradients, the computational overhead may be significant. However, the paper does not include any runtime or efficiency analysis, which is crucial for evaluating unlearning methods.

- The writing quality and clarity are fair, with several grammatical and stylistic errors. For example:
  - Line 34: “... is the case were users may ask ...” → should be “where” instead of “were”.
  - Line 47: “ASR foundation model” — ASR should be written in full at its first appearance.
  - “The key observation of this paper is that leading unlearning methods average over the retain set.” — this statement is vague and not a real observation.

**Questions:**

- The authors should clarify the core contribution of OrthoGrad. What specific challenges does gradient orthogonalization address, and what unique empirical phenomena support this choice?

- What is the computational complexity of OrthoGrad compared to retraining from scratch? How much efficiency improvement is achieved?

- Since OrthoGrad requires iterative parameter updates, how is the stopping criterion determined? In other words, how do the authors decide when the unlearning process is complete?

---

### Official Review · Reviewer_75JY · 2025-10-29

**Soundness:** 2
**Presentation:** 2
**Contribution:** 2
**Rating:** 4
**Confidence:** 4

**Summary:**

This paper proposes a machine unlearning method with access to a limited retain set. Specifically, the proposed method, OrthoGrad, projects the unlearning gradient onto the subspace orthogonal to the per-sample gradients of the retain set, and uses LoRA to update for unlearning, thereby conducting unlearning while maintaining the model utility.

Experiments are conducted across image classification and automatic speech recognition, and the machine unlearning methods are evaluated under sample-wise and class-wise unlearning settings. The paper also presents a new metric, Unlearning Impact Score (UIS) which measures how close the performance on the unlearning set is to that on the test set and the relative change in test accuracy, for comparison. Results show the effectiveness of the proposed method.

**Strengths:**

- The paper is easy to understand, and the motivation is clear. Without access to the full training set is practical in real-world scenarios.
- The proposed method is simple and makes sense to some extent.

**Weaknesses:**

- The paper aims for the performance on the unlearn set to match the original models' performance on the test set, which is confusing. If performance on the test set typically represents generalization, then a match between performance on the unlearning set and that on the test set indicates that effective forgetting is ambiguous, since this might indicate poor model generalization.

-  The paper claims that, different from the previous method GDR-GMA, which projects gradients based on averaged gradients, the proposed method instead considers per-sample gradients, and can achieve a stronger performance based on the experimental results. This claim is too strong and lacks a theoretical analysis to support it.

- Details of the proposed method are unclear. The proposed method uses LoRA, however, it is not clear which part of the network the LoRA is added.

**Questions:**

- It would be better to explain why unlearning should aim for a match between performance on the test set and that on the unlearning set.
- It would be better to detail the method, including the architecture.
- Since the projection relies on the gradients of the unlearning set and per-sample gradients of the retain set, what if some data points dominate the gradients? In Table 1, the variance is very high when considering the average of gradients of the retain set. Would this happen for that of the unlearning set as well?
- In line 295, 10% from the unlearn set is randomly selected to evaluate the model on the unlearned speaker; it would be better to explain why not evaluate on the whole unlearned set.
- In line 330, the method using the averaging gradients version is unstable, which could be due to that the method works well with some speakers but performs poorly on others. It would be better to further investigate why this could happen.
- In Table 2, SCRUB achieves a 100 WER, it would be better to further explain this result.
- Could the authors provide the reason to choose QR decomposition instead of something like SVD?

---

### Official Review · Reviewer_gFjg · 2025-10-30

**Soundness:** 2
**Presentation:** 4
**Contribution:** 2
**Rating:** 2
**Confidence:** 4

**Summary:**

The paper tackles machine unlearning when only a small retain set is available (a realistic setting for released foundation models whose original training data are inaccessible). It proposes OrthoGrad, which updates the model by projecting the unlearn gradient onto the subspace orthogonal to the per-sample gradients of a retain mini-batch—so that forgetting steps minimally interfere with what should be preserved. A geometric argument motivates restricting updates to the manifold that leaves retain losses unchanged, whose tangent space is the null space of per-sample retain gradients. Practically, the method forms a per-sample retain gradient matrix and uses a QR basis to subtract components from the unlearn gradient; the final update interpolates between the averaged retain gradient and the orthogonalized unlearn gradient, optionally with LoRA adapters. Experiments span image classification (ImageNet, ResNet-18/ViT), a proxy-retain scenario (CINIC-10/CIFAR-10), and ASR speaker unlearning with Whisper-Tiny, showing improved test performance at comparable forgetting versus recent baselines. The authors also introduce UIS, an aggregate metric intended to summarize retain utility + unlearning success.

**Strengths:**

1. The derivation that "retain-preserving directions = null space of per-sample retain gradients" is crisp and aligns well with the proposed projection operator, offering a principled lens on gradient interference in unlearning.
2. The paper addresses a practically important, under-served setting (retain data scarce or proxy), with protocol details for proxy-retain that are easy to reproduce.
3. Beyond standard image classification, the ASR speaker-forgetting benchmark is a valuable addition; here OrthoGrad matches forgetting while preserving test WER markedly better than several baselines.
4. Per-sample vs. mean retain gradients, with/without LoRA, and adding the retain gradient term are examined and show stability/variance benefits from per-sample orthogonalization.
5. The retain-size sweep suggests OrthoGrad degrades less across sizes than several baselines, consistent with the method’s design goal.

**Weaknesses:**

1. While the paper engages with multi-task conflict literature at a high level, it does not adequately situate OrthoGrad within recent unlearning works that manipulate retain/forget gradients via orthogonalization, surgery, multi-objective, or meta-learning, e.g., gradient projection/surgery and bargaining-style updates across [1-9]. The Related Work section briefly contrasts GDR-GMA but largely frames the distinction as "per-sample vs. batch-mean," which risks understating overlap with the broader family of gradient-space interventions. It would be helpful to add a dedicated subsection comparing OrthoGrad to gradient-orthogonalization/projection and gradient-conflict unlearning methods [1-9], clarifying what new theoretical/empirical insights per-sample subspace control brings over prior projections (what failure modes of mean-based projections does it fix?).

2. The method section states that per-sample retain gradients "can be obtained in a single forward-backward pass," which is generally not true for vanilla PyTorch autograd; naïve implementations require either looping vector-Jacobian products or specialized libraries (e.g., functorch/vmap) to materialize per-sample parameter-space gradients efficiently. As written, the claim could be misleading for large batches/models. It would be helpful to correct the statement to “single forward pass + structured backward(s)” or cite concrete tooling (e.g., per-sample gradient engines / micro-batching tricks), memory-time trade-offs or actual wall-clock profiling. The vanilla per-sample gradient computation as show below seems to be highly inefficient for large batch-sizes.

```
X = torch.randn(B, 10, requires_grad=True)
Y = torch.randn(B, 1)

out = model(X)
loss_per_sample = ((out - Y)**2).mean(dim=1)      # shape [B], scalar per sample
# Backprop a vector-Jacobian product for each sample:
grads_wrt_x = []
for i in range(B):
    g = torch.autograd.grad(loss_per_sample[i], X, retain_graph=True)[0][i].clone()
    grads_wrt_x.append(g)
grads_wrt_x = torch.stack(grads_wrt_x)            # [B, 10]
```
3. The paper attributes GDR-GMA’s lower performance to averaging retain gradients, but per-sample gradients are computationally heavier (based on the vanilla gradient computation procedure). The results show OrthoGrad’s test utility advantage, yet the paper does not quantify runtime/memory differences or discuss when mean-based methods may be preferable. A fair picture should acknowledge that OrthoGrad gains performance at extra compute or provide more clarity of trade-off between GDR-GMA and OrthoGrad. One way to go about it could be to report end-to-end throughput (examples/sec), peak memory, and backward counts for OrthoGrad vs. GDR-GMA across batch sizes and model scales (ResNet-18, ViT, Whisper-Tiny).


4. The propose metric to qunantity the robustness, i.e., UIS (especially the second term) pulls the unlearn-set performance toward the pretrained model’s test accuracy; this bakes in the assumption that "ideal forgetting should match test acc," which is not generally appropriate (e.g., if only 10% of training is unlearned and we truly unlearn them all, the unlearn accuracy could be very high and should not be penalized for deviating positively from test accuracy). The metric can therefore punish successful forgetting and conflate distributional issues. I would suggest the UIS or supplement with robust privacy-oriented metrics: membership-inference attack (MIA) AUC/advantage, point-based influence leakage, agreement with retrain-from-scratch (gold unlearning), following [2, 4, 8]. The paper discusses MIAs in the appendix, these should be included to main results with thorough analysis.


Current lean:
While I appreciate the paper’s clear geometric framing and the clean, practical recipe for per-sample gradient orthogonalization, my current lean is a reject because the methodological contribution over prior gradient-space unlearning, especially GDR-GMA [2] and [1], which already projects conflicting gradients and rescales updates, feels incremental: the main distinction emphasized is per-sample vs. batch-mean retain gradients, rather than a substantively new optimization principle or objective, and this contrast is already acknowledged in the paper (the paper notes GDR-GMA works on batch-level averages, whereas OrthoGrad uses per-sample spans).  At the evaluation level, the proposed UIS metric for classification encodes a debatable target by pulling both test accuracy and unlearn-set performance toward the pretrained model’s test accuracy (and early stopping is keyed to hitting a target accuracy), which can unintentionally penalize strong forgetting on small unlearn subsets and blur privacy/forgetting vs. utility, so UIS alone isn’t a robust arbiter of success without complementary privacy metrics (e.g., MIAs) and an holistic evaluation with respect to the gold-standard unlearning (Retrain).


[1] Learn to Unlearn for Deep Neural Networks: Minimizing Unlearning Interference with Gradient Projection, WACV 2024.

[2] GDR-GMA: Machine Unlearning via Direction-Rectified and Magnitude-Adjusted Gradients, ACMMM 2024.

[3] Learning to Unlearn for Robust Machine Unlearning, ECCV 2024.

[4] Scissorhands: Scrub Data Influence via Connection Sensitivity in Networks, ECCV 2024.

[5] Boosting alignment for post-unlearning text-to-image generative models, NeurIPS 2024.

[6] MUNBa: Machine Unlearning via Nash Bargaining, ICCV 2025.

[7] Meta-Unlearning on Diffusion Models: Preventing Relearning Unlearned Concepts, ICCV 2025.

[8] Learning to Unlearn while Retaining: Combating Gradient Conflicts in Machine Unlearning, ICCV 2025.

[9] GRU: Mitigating the Trade-off between Unlearning and Retention for LLMs, ICML 2025

**Questions:**

1. From lines 053-057 from the introduction, it is unlearn how the authors made the hypothesis of analyzing the gradient subspace. Why not identify the key parameter subspace that is most salient for the retain set and make updates to the orthogonal to that parameter subspace. It would be helpful to provide a the hypothesis of operating directly on the gradient subspace, and not any other alternate.

2. How exactly are per-sample parameter gradients computed at scale efficiently? Can you share runtime/memory vs. GDR-GMA?

3. Does OrthoGrad’s advantage persist when retain batches are larger and mean-based subspaces approximate per-sample spans? (An experiment could vary k while holding compute fixed.

---

### Author Response · Authors · 2025-11-17
**General Response**

We highlight several key differences and contributions of this work. Our method, OrthoGrad, introduces a new approach to machine unlearning by projecting the unlearning update onto the subspace orthogonal to all per-sample retain gradients.

OrthoGrad is fundamentally different from GDR-GMA. GDR-GMA projects the unlearn gradient against the average retain gradient, which ignores sample-level variation. In contrast, OrthoGrad computes the full set of per-sample retain gradients, performs a QR decomposition to extract an orthonormal basis, and then removes every component of the unlearn gradient that aligns with any retain direction. This ensures zero interference with the entire retain subspace, which is especially important when only limited retain data is available.

We acknowledge the additional computation introduced by per-sample gradients and QR decomposition. This limitation is discussed in the paper. Still, modern frameworks make these operations efficient; with vectorized backward passes, the overhead is roughly 2× compared to computing a mean gradient (e.g. using pytorch's vmap), even for large models such as Whisper.

Finally, our experiments show that OrthoGrad outperforms existing unlearning methods across multiple settings, architectures, and datasets, demonstrating its clear advantage in practical machine unlearning scenarios.

---

### Note · Authors · 2025-11-17

I have read and agree with the venue's withdrawal policy on behalf of myself and my co-authors.